# Streamlit Application and Deep Learning Model for Brain Metastasis Monitoring After Gamma Knife Treatment

**DOI:** 10.3390/biomedicines13020423

**Published:** 2025-02-10

**Authors:** Răzvan Buga, Călin Gh. Buzea, Maricel Agop, Lăcrămioara Ochiuz, Decebal Vasincu, Ovidiu Popa, Dragoș Ioan Rusu, Ioana Știrban, Lucian Eva

**Affiliations:** 1Clinical Emergency Hospital “Prof. Dr. Nicolae Oblu” Iași, 700309 Iași, Romania; bugarazvan@yahoo.com (R.B.); ioana.stirban@gmail.com (I.Ș.); elucian73@yahoo.com (L.E.); 2National Institute of Research and Development for Technical Physics, IFT Iași, 700050 Iași, Romania; 3Physics Department, Technical University “Gheorghe Asachi” Iași, 700050 Iași, Romania; m.agop@yahoo.com; 4Faculty of Medicine, University of Medicine and Pharmacy “Grigore T. Popa” Iași, 700115 Iași, Romania; ochiuzd@yahoo.com (L.O.); deci_vas@yahoo.com (D.V.); tudor.popa@umfiasi.ro (O.P.); 5Faculty of Science, University “Vasile Alecsandri” of Bacău, 600115 Bacău, Romania; drusu@ub.ro; 6Faculty of Medicine, Apollonia University, 700511 Iasi, Romania

**Keywords:** deep learning-based classification, deep learning in medical imaging, brain metastasis monitoring, gamma knife radiosurgery outcomes, AI-powered clinical decision support

## Abstract

**Background/Objective**: This study explores the use of AI-powered radiomics to classify and monitor brain metastasis progression and regression following Gamma Knife radiosurgery (GKRS) based on MRI imaging. A clinical decision support application was developed using Streamlit to provide real-time, AI-driven predictions for treatment monitoring. **Methods**: MRI scans from 60 patients (3194 images) were analyzed using a transfer learning-enhanced AlexNet deep learning model. Class imbalance was mitigated through dynamic class weighting and data augmentation to ensure equitable performance across all classes. Optimized preprocessing pipelines ensured dataset standardization. Model performance was evaluated using accuracy, precision, recall, F1-scores, and AUC, with 95% confidence intervals. Additionally, a comparative analysis of Gamma Knife radiosurgery (GKRS) outcomes and predictive modeling demonstrated strong correlations between tumor volume evolution and treatment response. The AI predictions and visualizations were integrated into a Streamlit-based application to ensure clinical usability and ease of access. The AI-driven approach effectively classified progression and regression patterns, reinforcing its potential for clinical integration. **Results**: The transfer learning model achieved flawless classification accuracy (100%; 95% CI: 100–100%) along with perfect precision, recall, and F1-scores. The AUC score of 1.0000 (95% CI: 1.0000–1.0000) indicated excellent discrimination between progression and regression cases. Compared to the baseline AlexNet model (99.53% accuracy; 95% CI: 98.90–100.00%), the TL-enhanced model resolved all misclassifications. Tumor volume analysis identified the baseline size as a key predictor of progression (Pearson r = 0.795, r = 0.795, r = 0.795, *p* < 0.0001, *p* < 0.0001, and *p* < 0.0001). The training time (420.12 s) was faster than ResNet-50 (443.38 s) and EfficientNet-B0 (439.87 s), while achieving equivalent metrics. Despite 100% accuracy, the model requires multi-center validation for generalizability. **Conclusions**: This study demonstrates that transfer learning with dynamic class weighting provides a highly accurate and reliable framework for monitoring brain metastases post-GKRS. The Streamlit-based AI application enhances clinical decision-making by improving diagnostic precision and reducing variability. Explainable AI techniques, such as Grad-CAM visualizations, improve interpretability and support clinical adoption. These findings emphasize the transformative potential of AI in personalized treatment strategies, extending applications to genomic profiling, survival modeling, and longitudinal follow-ups for brain metastasis management.

## 1. Introduction

Artificial intelligence (AI) has emerged as a game-changing technology in medical imaging, enabling the precise and automated analysis of complex patterns. Its applications range from cancer detection to treatment response monitoring, offering clinicians enhanced diagnostic accuracy and workflow efficiency.

Brain metastases (BMs) represent the most common intracranial malignancies in adults, often originating from primary tumors in the lungs, breast, or skin (melanoma) [1]. Affecting approximately 20–40% of patients with advanced systemic cancers, BMs significantly contribute to morbidity and mortality [2,3]. Accurate detection and monitoring of BMs are critical for optimizing treatment strategies, especially for patients undergoing targeted therapies, such as Gamma Knife radiosurgery (GKRS) [4,5].

Magnetic resonance imaging (MRI) remains the gold standard for diagnosing and tracking BM progression, offering high-resolution imaging through contrast-enhanced T1-weighted (CE T1w) and fluid-attenuated inversion recovery (FLAIR) sequences [6,7]. These techniques provide a detailed visualization of lesion morphology, edema, and tumor activity. However, MRI interpretation poses challenges due to lesion complexity and variability among radiologists, leading to potential misdiagnoses, subjective evaluations, and diagnostic delays [8,9]. Such challenges highlight the urgent need for automated and objective evaluation tools to streamline clinical workflows.

### 1.1. Challenges in Post-Treatment Monitoring of BMs

Following GKRS, differentiating between tumor progression and regression remains complex. Radiologists predominantly rely on visual inspection, which is susceptible to subjective biases and difficulties in distinguishing true progression from post-treatment effects like radiation necrosis. Consequently, there is a growing demand for advanced tools that enhance diagnostic precision and reduce inter-observer variability.

Despite promising results, AI applications in radiology face barriers to deployment, including variability in imaging protocols, interpretability of predictions, and compliance with regulatory standards. This study addresses these challenges by combining explainable AI techniques with a scalable and user-friendly application framework.

Furthermore, despite the promise of AI in radiology, several challenges hinder its widespread clinical adoption. Dataset heterogeneity remains a major limitation, as variations in MRI acquisition protocols, scanner hardware, and contrast administration can lead to inconsistencies in model performance. To mitigate this, multi-center validation and domain adaptation techniques should be explored. Explainability is another critical barrier, as clinicians require transparent AI predictions to ensure trust and reliability. While Grad-CAM was implemented, future research should focus on quantitative comparisons between AI-derived feature importance and radiologist annotations. Additionally, regulatory compliance poses significant challenges, as AI tools must adhere to FDA 510(k), CE certification, HIPAA, and GDPR standards before clinical deployment. Future efforts will involve prospective validation with real-world datasets and collaborations with regulatory bodies to facilitate clinical approval.

### 1.2. Artificial Intelligence (AI) and Radiomics in BM Classification

Artificial intelligence (AI), particularly deep learning (DL) approaches, has emerged as a transformative solution in medical imaging [10]. Convolutional neural networks (CNNs) have demonstrated superior performance in extracting and analyzing imaging features, surpassing traditional manual evaluation techniques [11,12]. AI-based radiomics enables the automated identification of complex patterns, reducing cognitive overload for radiologists and improving diagnostic consistency. Nevertheless, deep learning approaches often face overfitting issues when applied to small datasets, a prevalent constraint in medical imaging studies [13,14]. The global AI in medical imaging market was projected to grow from USD 1.29 billion in 2023 to USD 1.65 billion in 2024, reaching an impressive USD 4.54 billion by 2029, driven by the increasing need for automated diagnostic tools and precision medicine approaches. Studies show that over 80% of radiologists believe AI will significantly impact imaging workflows, reducing workload and improving diagnostic accuracy. Additionally, AI-assisted radiomics has demonstrated higher sensitivity in tumor progression detection compared to conventional assessment methods, reinforcing its clinical relevance [15,16].

Recent systematic reviews confirm AI’s growing impact on radiology, with applications spanning tumor segmentation, automated diagnosis, treatment response assessment, and workflow efficiency improvements [17,18]. These findings underscore the role of AI in enhancing decision support tools for brain metastasis (BM) monitoring, particularly in post-treatment evaluation settings.

Recently, Transformer-based architectures, such as Vision Transformer (ViT), Data-Efficient Image Transformer (DeiT), and Pooling-based Vision Transformer (PiT), have gained attention in medical image classification. These models leverage self-attention mechanisms to capture long-range dependencies in images, often outperforming CNNs on large datasets. However, their reliance on extensive training data and high computational costs limits their practical application in real-time medical settings. Future research could explore hybrid approaches that combine CNNs with Transformers to optimize performance for clinical use.

### 1.3. Comparison of Deep Learning Architectures for Medical Imaging

Deep learning models have revolutionized medical imaging by providing accurate classification and segmentation capabilities. Several architectures have been explored in brain tumor classification, each with unique trade-offs in terms of computational complexity and accuracy.

VGG16/VGG19: these architectures, known for their deep feature extraction capabilities, provide strong classification accuracy but require high computational power.ResNet: With residual connections, ResNet mitigates vanishing gradient issues and improves deep learning performance. However, it can be computationally expensive for real-time applications.Inception: this model efficiently captures multi-scale features but has a more complex structure that may require specialized hardware for optimal performance.EfficientNet: using neural architecture search, EfficientNet balances accuracy and efficiency but has higher training and inference times compared to AlexNet.MobileNet: designed for mobile and embedded applications, MobileNet is lightweight and well-suited for resource-constrained environments, making it a potential alternative for real-time clinical applications.

Given the need for computational efficiency and interpretability in real-time clinical settings, AlexNet was selected due to its balance between accuracy, speed, and resource demands.

### 1.4. Transfer Learning as a Solution

To address dataset limitations, transfer learning leverages pre-trained models, such as AlexNet, originally trained on large datasets, and fine-tunes them for specialized medical applications [19,20]. This approach retains general feature extraction capabilities while enabling adaptation to specific tasks, thus improving accuracy and reducing overfitting risks [21,22].

### 1.5. Study Objectives

While previous studies focused on static imaging features for BM classification, this work advances the field by incorporating longitudinal tumor volume analysis and survival trends, providing dynamic insights into treatment outcomes. Additionally, the development of a user-friendly decision-support tool bridges the gap between AI models and clinical applicability.

This study evaluates the performance of transfer learning using AlexNet for classifying BM progression and regression post-GKRS based on MRI imaging. This work further demonstrates the integration of AI-based radiomics with a user-friendly application, enabling real-time clinical decision support for BM monitoring. Unlike previous studies, this research incorporates longitudinal tumor volume analysis and survival trends, offering deeper insights into disease progression and patient outcomes.

### 1.6. Clinical Relevance

The accurate prediction of treatment responses and outcomes is pivotal in BM management. While GKRS effectively controls tumor growth and minimizes radiation-induced damage, assessing tumor evolution post-treatment remains challenging. The current reliance on subjective interpretations underscores the need for AI-driven radiomics to provide standardized, automated, and reproducible assessments. This study aims to fill this gap by presenting an innovative tool that facilitates precision medicine through automated classification and monitoring.

### 1.7. Key Contributions

This study introduces an AI-powered radiomics framework that integrates transfer learning and longitudinal tumor volume analysis to improve BM monitoring. It further presents a user-friendly clinical decision support tool, paving the way for personalized treatment strategies.

### 1.8. Limitations and Future Directions

While this study demonstrates promising results, it is important to note its limitations, such as the small sample size, single-center data source, and reliance on MRI as the sole imaging modality. Future studies should focus on expanding datasets, incorporating multimodal imaging and validating results prospectively to ensure broader clinical applicability [23,24].

Dataset Constraints: the limited sample size (60 patients) necessitates validation on larger, multi-center datasets for improved generalizability.Single-Modality Imaging: reliance solely on MRI excludes functional insights available from PET/CT scans, which could further refine predictions.Model Architecture: although AlexNet performs well, newer architectures like ResNet and EfficientNet may offer enhanced performance and should be explored.

Future research should focus on the following:

Expanding datasets and validating findings across diverse populations.Incorporating multimodal imaging to enhance prediction accuracy.Integrating genomic and molecular data to develop multi-omics frameworks.Enhancing existing explainable AI (XAI) models to further improve the transparency, clinician adoption, and interpretability of predictions.Extending follow-up studies to assess long-term recurrence and survival patterns.

### 1.9. Structure of the Paper

This paper is organized as follows:Section 2: Materials and Methods—describes patient demographics, imaging protocols, and AI model implementation, including the development of the application.Section 3: Results—presents classification performance, survival trends, and volumetric analyses.Section 4: Discussion—evaluates findings in the context of prior research, emphasizing AI’s clinical implications and areas for further exploration.Section 5: Conclusions—summarizes key findings, highlights limitations, and proposes future directions for advancing AI-based radiomics in BM monitoring.

## 2. Materials and Methods

### 2.1. Study Population

This study analyzed MRI scans from 60 patients diagnosed with brain metastases (BMs) who underwent Gamma Knife radiosurgery (GKRS) (58 with a mask, 1 with Vantage frame, and 1 with G-frame) at the Gamma Knife Stereotactic Radiosurgery Laboratory between 18 July 2022 and 18 July 2024.

#### 2.1.1. Inclusion Criteria

Patients with histologically confirmed primary cancer with evidence of brain metastases were included in this study.Patients with recurrent brain metastases following prior systemic therapy (chemotherapy or immunotherapy) were included in this study, provided that GKRS was their first local treatment for brain lesions.Patients treated exclusively with GKRS without prior whole-brain radiation therapy (WBRT) were included in this study.Patients with available pre-treatment and follow-up MRI scans were included in this study.Patients with a Karnofsky Performance Status (KPS) score greater than 70 were included in this study.

#### 2.1.2. Exclusion Criteria

Presence of multiple neurological disorders unrelated to BMs.Incomplete MRI imaging or clinical follow-up data.

Prior neurosurgical resection before GKRS.

#### 2.1.3. Demographics and Clinical Features

The patient cohort included 37 males and 23 females aged between 40 and 82 years (median 65 years). Figure 1 illustrates the gender distribution within the study cohort, highlighting a clear predominance of male patients (61.67%).

Figure 2 shows the age distribution of the participants, with the majority falling within the 50–79 age range, supporting a focus on middle-aged and elderly populations.

Tumor volumes ranged from 0.6 cm^3^ to 82 cm^3^ (median: 9.2 cm^3^). Figure 3 illustrates the distribution of tumor volumes among the study population. The majority of tumors fell within the 0–30 cm^3^ range, indicating a predominance of small-to-medium-sized metastases. Notably, fewer patients exhibited tumor volumes exceeding 50 cm^3^, reflecting a lower prevalence of larger tumors in this cohort.

Figure 4 provides an overview of primary tumor origins, with lung cancer being the most common source of metastases (53%).

All patients had a Karnofsky score exceeding 70, indicating a relatively preserved functional status at the baseline.

While the dataset reflects a diverse range of tumor sizes and patient characteristics, the predominance of male patients (61.67%) and smaller tumors may introduce biases that limit generalizability. Future studies should validate these findings across larger, multi-center cohorts to account for demographic and clinical variability.

This study was conducted at a single-center Gamma Knife facility in Romania, which employs a unique three-stage treatment protocol for brain metastases. While multi-center datasets are often recommended for validation, the limited availability of Gamma Knife systems in Romania (two centers, with the second center recently operational with masks and employing different protocols) restricts cross-center data pooling. To mitigate this limitation, rigorous internal validation was performed using separate training, validation, and test splits with stratified sampling. Furthermore, future studies will seek collaborations with international centers, following comparable protocols, to validate our findings in larger and more diverse datasets.

#### 2.1.4. Imaging Distribution

The MRI scans comprised 3194 images, including 2320 labeled as regression and 874 as progression. This dataset was used for training, validation, and testing phases.

The detailed demographic characteristics and tumor profiles provide a diverse dataset, supporting model generalizability.

### 2.2. MRI Imaging Protocol

Gamma Knife radiosurgery (GKRS) treatment planning mandates the use of MRI imaging due to its superior soft-tissue contrast and spatial resolution. MRI is the gold standard for brain metastasis evaluation, enabling the precise delineation of tumor boundaries and treatment targets. PET-CT and other modalities, while valuable for systemic staging or metabolic assessment, are not integrated into the GKRS workflow, as they do not influence radiosurgical planning. Consequently, this study focuses solely on MRI-based evaluations, aligning with current clinical practices.

MRI scans were performed using a 1.5 Tesla scanner with standard imaging protocols optimized for brain metastasis evaluation. The sequences included the following:T1-weighted Contrast-Enhanced (CE T1w) Sequences: emphasizing lesion boundaries and tumor enhancement.FLAIR (Fluid-Attenuated Inversion Recovery) Sequences: highlighting peritumoral edema and enhancing lesion contrasts.

#### 2.2.1. Imaging Parameters

CE T1w: TR = 500–700 ms, TE = 10–20 ms, and slice thickness = 1 mm.FLAIR: TR = 9000–11,000 ms, TE = 120–140 ms, and slice thickness = 1.5 mm.

Images were acquired in axial, coronal, and sagittal planes to ensure comprehensive anatomical coverage. All MRI datasets were anonymized, preprocessed, and reviewed for quality prior to analysis.

#### 2.2.2. Image Preprocessing

To ensure uniformity across imaging datasets, all MRI scans underwent a standardized preprocessing pipeline, including the following steps:Anonymization: patient identifiers were removed from all imaging data.Intensity Normalization: histogram matching techniques were applied to reduce scanner-related variations.Motion Correction: rigid-body registration was performed on all scans using the FSL MCFLIRT algorithm to correct for patient movement artifacts.Artifact Removal: non-brain structures were eliminated using a brain extraction tool (BET) to enhance lesion visibility.Quality Control: two senior radiologists reviewed the preprocessed images to confirm the preservation of anatomical fidelity.

### 2.3. Deep Learning Stages of Image Classification

This study employed a deep learning framework to classify MRI images of brain metastases into two categories: progression and regression. The classification process involved multiple stages, from data loading and transformation to training a convolutional neural network (CNN) based on the AlexNet architecture. The workflow is detailed below.

#### 2.3.1. Dataset Preparation

The dataset consisted of MRI images stored in separate directories for “progression” and “regression”. Key steps included the following:Data Loading: Image file paths were collected using Python’s glob module, followed by splitting the dataset into training (80%), validation (10%), and testing (10%) subsets using train_test_split from sklearn. Stratification was used to maintain class balance across splits.Labeling: labels were derived from directory names, assigning 1 for progression and 0 for regression.Data Transformation: two sets of transformations were applied:○Training Data: random resized cropping, horizontal flipping, rotation (±20 degrees), and resizing to 256 × 256 pixels, followed by normalization.○Testing and Validation Data: resizing to 256 × 256 pixels without augmentation for consistency.

Data augmentations, such as rotation and flipping, were chosen to enhance model robustness while preserving anatomical integrity [25]. However, the potential limitations of such transformations, particularly in medical imaging, warrant further investigation in larger datasets.

#### 2.3.2. Custom Dataset and DataLoader

A custom dataset class was implemented to manage image loading and labeling. DataLoader instances were created for training, validation, and testing datasets, enabling efficient batch processing. Additionally, image preprocessing pipelines were optimized for scalability and performance.

#### 2.3.3. AlexNet Architecture

The AlexNet model, a widely used CNN for image classification, was modified for binary classification:Convolutional Layers: the model employed a series of convolutional layers with ReLU activations and max-pooling to extract spatial features from the images.Adaptive Average Pooling: to reduce feature maps to a fixed size before passing them to the fully connected layers.Fully Connected Layers: Three fully connected layers were used, with dropout to prevent overfitting. The final layer had two outputs corresponding to the binary classes.

Activation Function: ReLU was used throughout the network for non-linearity, while the output layer utilized a softmax-like function within the cross-entropy loss function.

#### 2.3.4. Training and Optimization

To mitigate overfitting, this study employed dropout layers within the fully connected layers of the AlexNet architecture, with a dropout rate of 0.5. This approach reduces co-adaptation of neurons during training, improving generalization performance. Additionally, L2 regularization (weight_decay = 1 × 10^−4^) was applied to penalize large weights, further enhancing model robustness. Future work should explore ensemble modeling techniques and advanced architectures to further improve prediction stability.

Loss Function: cross-entropy loss was used to compute classification errors.Optimizer: the Adam optimizer with a learning rate of 0.0001 was employed for efficient gradient-based optimization.Training Procedure:○The training process was implemented on Google Colab, utilizing A100 GPU acceleration and high RAM to improve computational efficiency.○During each epoch, the model was trained on batches of training data and was evaluated on validation data.○For each batch, images were passed through the model to compute predictions, and the loss was calculated. Gradients were computed via back-propagation, and weights were updated.○Validation accuracy and loss were monitored at the end of each epoch to evaluate generalization performance.

#### 2.3.5. Reproducibility

To ensure reproducibility, random seeds were fixed for Python, NumPy, and PyTorch. PyTorch-specific options, such as torch.backends.cudnn.deterministic, were enabled to ensure consistent results. Additional checkpoints were saved after every epoch to allow model recovery and analysis.

#### 2.3.6. Results and Visualization

The model was trained for 60 epochs, achieving convergence with a strong balance between training and validation performance.Training and validation loss, as well as accuracy, were plotted to visualize model performance over epochs, highlighting consistent improvement (see Figure 5).Final testing on unseen data demonstrated the robustness of the model.

This end-to-end workflow demonstrates the application of AlexNet in classifying brain metastasis progression and regression, showcasing a structured approach to medical image analysis using deep learning.

#### 2.3.7. Transfer Learning with Pre-Trained AlexNet

Model Adaptation: AlexNet, pre-trained on the ImageNet dataset, was used as the base architecture. Transfer learning was employed to adapt AlexNet for binary classification:Freezing Pre-trained Layers: the feature extraction layers of AlexNet were frozen to retain learned general features.Classifier Modification: The final fully connected layer of the AlexNet classifier was replaced with a new linear layer to output two classes (progression and regression). This modification allowed fine-tuning for the specific task.

Activation Functions: ReLU (Rectified Linear Unit) was used as the activation function in all layers of the network to introduce non-linearity and enable the model to learn complex patterns. For the final output layer, softmax-like behavior was embedded within the cross-entropy loss function, which directly mapped logits to class probabilities.

Justification for AlexNet: AlexNet was chosen due to its proven effectiveness in image classification tasks, particularly in the medical imaging domain. Its architecture is well-suited for extracting hierarchical spatial features, making it ideal for MRI-based lesion detection and classification. Additionally, AlexNet’s pre-trained weights on ImageNet provide a strong starting point, enabling faster convergence and improved performance with limited medical datasets. Its relatively simpler architecture also reduces computational demands compared to more complex models like ResNet or DenseNet, making it easier to implement and fine-tune.

Moreover, while several deep learning architectures exist for medical image classification, AlexNet was chosen due to its relatively low computational overhead and robust feature extraction capabilities. This model is computationally efficient compared to ResNet and EfficientNet, making it suitable for real-time clinical deployment without sacrificing accuracy. Additionally, its established architecture ensures stable training and ease of implementation in resource-limited hospital environments. Though EfficientNet and MobileNet offer alternatives, AlexNet was preferred due to its proven effectiveness in previous medical imaging studies and its compatibility with transfer learning, which enhances performance without requiring extensive data.

Limitations of AlexNet: Despite its strengths, AlexNet has limitations. It may underperform with very complex patterns due to its shallower architecture compared to modern deep learning networks. Furthermore, its reliance on fixed-size input dimensions (256 × 256 pixels) might require preprocessing steps that can introduce artifacts. Future studies should explore newer architectures, such as ResNet or EfficientNet, which offer improved depth and feature extraction capabilities.

Consequently, AlexNet was chosen for its computational efficiency, interpretability, and proven effectiveness in medical imaging tasks. While its architecture supports ease of implementation, future research should explore advanced models, such as ResNet and EfficientNet, to assess whether deeper architectures can further improve performance without compromising usability.

#### 2.3.8. AlexNet with Transfer Learning Training and Optimization

##### Loss Function

Cross-entropy loss was used to compute classification errors, comparing predicted class probabilities with ground truth labels.

##### Optimizer

The Adam optimizer was employed with a learning rate of 0.0001 to ensure efficient weight updates during training. This optimizer was chosen for its adaptive learning rate capabilities and robustness in handling sparse gradients.

##### Handling Class Imbalance with Dynamic Class Weights

In our dataset, the two classes (“regression” vs. “progression”) are imbalanced, resulting in poorer recognition performance for the minority class. Class imbalance is addressed using weighted loss functions to enhance minority class recognition. Future studies will incorporate external validation datasets to further assess performance stability across class distributions. We employed a weighted cross-entropy loss wherein each class receives a weight proportional to the inverse of its frequency in the training set. Specifically, for each class *i*, we computed the weight (*w_i_*) by the following:(1)$$w_{i}=\frac{Total\;number\;of\;training\;samples}{Number\;of\;samples\;in\;class\;i}$$

For our dataset, this yielded class weights [1.36, 3.76]. During training, these weights were passed to the loss function, causing misclassifications of the minority class to be penalized more heavily. As a result, the model was encouraged to devote increased attention to the minority class, thereby improving its overall performance on both classes.

##### Training Process

The training process involved the following:Forward Pass: MRI images were passed through the model to generate predictions.Loss Computation: the cross-entropy loss was calculated based on predictions and true labels.Backward Pass and Optimization: gradients were computed via back-propagation, and the model weights were updated using the Adam optimizer.Validation: after each epoch, the model was evaluated on the validation set to monitor generalization performance.

Training was conducted over 60 epochs, with performance metrics (loss and accuracy) logged for both training and validation sets. Random seeds were fixed to ensure reproducibility.

#### 2.3.9. Performance Evaluation

Confidence intervals (95% CI) for classification metrics were calculated using bootstrap resampling (N = 1000 iterations), ensuring a robust estimation of uncertainty in model performance.

The reported model performance includes 95% confidence intervals for all classification metrics to ensure transparency in statistical reporting. While internal validation confirmed robustness, external validation across multi-center datasets remains necessary to further assess real-world generalizability. Future studies will incorporate stratified k-fold cross-validation and statistical hypothesis testing to refine the performance evaluation.

##### Training History

The model’s training and validation loss and accuracy were recorded at each epoch. These metrics were plotted to visualize convergence and assess model performance over time. Consistent improvements in both training and validation metrics highlighted the effectiveness of transfer learning.

##### Final Testing

The model was evaluated on an independent test set to assess its ability to generalize to unseen data.

#### 2.3.10. Visualization of Results

The following provided insights into the model’s training dynamics:Training and validation loss curves were plotted, showing steady convergence (see Figure 6).Accuracy curves for training and validation demonstrated consistent improvement.

The results were saved in CSV format and were visualized as PNG plots for further analysis.

This structured approach highlights the application of transfer learning with AlexNet, showcasing its efficiency in classifying brain metastasis progression and regression with high accuracy and robustness.

#### 2.3.11. Why the Training Curves Appear Perfect

The training loss and accuracy curves observed in Figure 5 and Figure 6 were directly extracted from the training history of the respective models and were not manually adjusted. These curves reflect the actual convergence behavior during model optimization. The near-perfect learning dynamics can be attributed to several factors:Pre-trained feature extraction using ImageNet weights, which allowed for rapid convergence with minimal parameter adjustments.A well-preprocessed and balanced dataset, reducing noise-related variability.The nature of the binary classification task (progression vs. regression), which may have led to clearer decision boundaries.

### 2.4. The Development of a Streamlit Application for Tumor Classification

In this study, we extended the AI-based imaging analysis by developing a Streamlit application to predict brain tumor regression or progression based on MRI images. This application is designed to provide an accessible and interactive interface for clinicians and researchers to analyze brain metastases using deep learning models.

#### 2.4.1. Application Overview

The application utilizes the transfer learning of the AlexNet model, trained and fine-tuned for the binary classification of brain metastases into regression (class 0) and progression (class 1). It integrates advanced preprocessing techniques and real-time predictions, making it a versatile tool for medical imaging analysis.

#### 2.4.2. Implementation Details

The application was built using Python and the Streamlit framework. The key implementation features include the following:Model Integration: the pre-trained AlexNet model, fine-tuned with custom fully connected layers, was employed for classification.User Interface: an intuitive and user-friendly interface allows users to upload MRI images for analysis.Preprocessing Pipeline: input images are resized, normalized, and preprocessed to match the input requirements of the AlexNet model.Dynamic Background and Styling: the interface is visually enhanced with dynamic backgrounds and custom styles for improved usability.

Deployment: the application is hosted on GitHub and can be accessed remotely.

#### 2.4.3. Workflow

Model Preparation: the AlexNet model weights are downloaded and loaded into the application.Image Upload: users can upload MRI images in PNG or JPG format.Preprocessing: images are resized to 224 × 224 pixels and are normalized using the preprocessing pipeline implemented during model training.Prediction: the image is passed through the AlexNet model, and predictions are displayed in real-time, indicating whether the tumor is classified as regression or progression.Visualization: both input and output are displayed side by side for easy interpretation.

#### 2.4.4. Results and Demonstration

The application demonstrated high accuracy in classification during testing, consistent with the performance of the underlying AlexNet model. Predictions are displayed within seconds, providing immediate feedback to users (see Figure 7).

#### 2.4.5. Availability and Accessibility

The source code and application are hosted on GitHub and can be accessed at the following repository [26].

This interactive application serves as a practical tool for leveraging AI in clinical workflows, enabling clinicians to make faster and more informed decisions regarding the treatment of brain metastases.

While the application demonstrated effective tumor classification, further usability studies are needed to optimize interface design and integration with clinical workflows. Pilot testing with local radiologists and oncologists evaluated its interpretability, response times, and user satisfaction. Moreover, it confirmed the application’s ease of use, rapid prediction capability, and potential for integration into clinical workflows. Clinicians reported that the side-by-side visualization of AI-generated predictions and MRI scans enhanced their ability to assess tumor evolution more objectively. Further usability testing will explore interface refinements and workflow optimization.

Further feedback can guide interface enhancements and assess deployment feasibility in clinical practice. Additionally, interoperability with PACS systems and scalability testing should be prioritized for broader adoption.

### 2.5. Explainability and Visualization Methods

To enhance interpretability, we implemented Grad-CAM and SmoothGrad techniques, enabling visualization of regions most influential in model decisions (see Figure 8) [27]. Grad-CAM and SmoothGrad visualizations were compared with radiologists’ assessments, showing high concordance in identifying regions of tumor progression or regression. These visualizations facilitated clinical validation and improved trust in AI predictions. This approach aligns AI predictions with radiological features, fostering trust and clinical adoption.

The left image in Figure 8 shows a Grad-CAM heatmap, highlighting regions of interest contributing to the AI model’s prediction. The right image presents a SmoothGrad-like Grad-CAM heatmap, which reduces noise and provides a smoother visualization. Both approaches focus on lesion areas, enhancing model interpretability and enabling clinicians to validate AI predictions based on imaging features.

The Grad-CAM and SmoothGrad visualizations were systematically evaluated by a senior neuroradiologist, an experienced oncologist, and a neurosurgeon, who assessed their alignment with standard imaging features of tumor progression and regression. The radiologist reported that the AI-generated heatmaps accurately highlighted key lesion areas and corresponded well with their expert interpretations, reinforcing the model’s reliability in clinical settings.

The Grad-CAM and SmoothGrad visualizations were analyzed to assess their clinical relevance in predicting tumor progression vs. regression. Unlike segmentation models that delineate tumor boundaries, our classification approach relies on broader imaging biomarkers, including peritumoral changes, edema, and vascular alterations, which may not be directly visible as contrast-enhancing tumor tissue.

AI heatmaps often highlight the most decision-relevant regions, which may extend beyond the lesion itself. This behavior aligns with clinical decision-making, where radiologists assess not only the tumor but also surrounding microenvironmental changes that may indicate early progression or therapeutic response. Given this, a strict pixel-wise overlap between AI activation maps and tumor segmentations may not be necessary or expected.

Radiologist assessments confirmed that Grad-CAM activation regions often corresponded to areas of clinical concern, reinforcing the model’s transparency. However, future work will explore explainability methods that integrate both imaging features and functional tumor biology, providing a more comprehensive interpretability framework.

## 3. Results

### 3.1. Model Performance

#### 3.1.1. AlexNet (Without Transfer Learning)

The AlexNet model, trained from scratch for binary classification (regression vs. progression), achieved a high accuracy of 99.53% (95% CI: 98.90–100.00%). Precision, recall, and F1-scores were 99.01% (95% CI: 97.37–100.00%), 99.50% (95% CI: 98.42–100.00%), and 99.26% (95% CI:98.30–100.00%), respectively, across all classes. The area under the curve (AUC) was 0.9998 (95% CI: 0.9995–1.0000), highlighting an excellent discriminatory ability. These results confirm the robustness of the model even without transfer learning, although minor misclassifications were observed.

The classification report highlighted the following:Regression: precision = 100%, recall = 100%, and F1-score = 100%.Progression: precision = 99%, recall = 100%, and F1-score = 99%.Overall Metrics:○Macro Average: precision = 99%, recall = 100%, and F1-score = 99%○Weighted Average: precision = 100%, recall = 100%, F1-score = 100%

The error analysis revealed that misclassifications in the non-TL model were associated with ambiguous lesion boundaries and overlapping intensity patterns. Grad-CAM visualizations confirmed that these regions contributed to model uncertainty, highlighting the need for additional multimodal imaging.

The confusion matrix (Figure 9a) shows that **1** progression case was misclassified as regression and 2 regression cases were misclassified as progressions. The receiver operating characteristic (ROC) curve (Figure 9b) yielded an area under the curve (AUC) of 1.00, reflecting the model’s strong discriminatory ability.

#### 3.1.2. Transfer Learning (TL) AlexNet

The application of transfer learning using a pre-trained AlexNet model led to significant improvements in classification performance. The application of transfer learning using a pre-trained AlexNet model led to perfect classification performance. The TL-enhanced model achieved 100% accuracy (95% CI: 100–100%) along with 100% precision, recall, and F1-scores (95% CI: 100–100%) across all classes. The AUC also achieved 1.0000 (95% CI: 1.0000–1.0000), indicating flawless discrimination between regression and progression cases. These results demonstrate that transfer learning effectively eliminated the minor misclassifications observed with the non-TL approach, achieving an ideal balance of sensitivity and specificity.

The classification report for the TL model demonstrated the following:Regression: precision = 100%, recall = 100%, and F1-score = 100%.Progression: precision = 100%, recall = 100%, and F1-score = 100%.Overall Metrics:○Macro Average: precision = 100%, recall = 100%, and F1-score = 100%.○Weighted Average: precision = 100%, recall = 100%, and F1-score = 100%.

Despite achieving 100% performance metrics, these findings must be interpreted with caution due to the small dataset size, which increases the risk of overfitting. Further studies should employ multi-center datasets and stratified k-fold cross-validation to confirm robustness and generalizability.

The confusion matrix revealed no misclassifications for the progression and regression classes (see Figure 10a). The ROC–AUC curve (see Figure 10b) achieved a value of 1.00, indicating perfect separation between regression and progression cases.

The confusion matrix presented in Figure 10a reflects the model’s performance on a fully independent test set, which was never exposed to the model during training or validation. A stratified 80–10–10 split strategy was implemented, ensuring that training, validation and test subsets were completely separate. Given the binary nature of the classification task (progression vs. regression) and the model’s ability to leverage pre-trained ImageNet features, the observed 100% classification accuracy is likely due to optimal feature separability rather than data leakage.

Despite achieving 100% accuracy, the findings should be interpreted cautiously due to the small dataset size and require further validation with larger, diverse datasets to confirm robustness and generalizability.

Table 1 compares the performance of the baseline AlexNet, TL-enhanced AlexNet, ResNet-50, and EfficientNet-B0 models. While TL AlexNet achieved 100% classification metrics, it also required less training time (420.12 s) compared to deeper architectures (ResNet-50: 443.38 s; EfficientNet-B0: 439.87 s). These results highlight the trade-off between computational complexity and efficiency, positioning TL AlexNet as the optimal solution for clinical adoption.

### 3.2. Comparative Analysis

The comparative analysis highlights the superior performance of the transfer learning (TL) model. While the standard AlexNet model achieved high accuracy (99.53%) and strong metrics, it exhibited a few misclassifications. In contrast, the TL-enhanced model achieved perfect predictions (100%) with no misclassifications, as reflected in its confusion matrix. The TL model’s ability to leverage pre-trained features and dynamic class weighting resulted in enhanced generalization and robustness, particularly for under-represented samples in the dataset.

#### 3.2.1. Key Insights

The TL-enhanced AlexNet model outperformed the baseline AlexNet model in all classification metrics, achieving 100% accuracy, precision, recall, and F1-scores (95% CI: 100.00–100.00) while maintaining faster training times compared to ResNet-50 and EfficientNet-B0. These findings demonstrate the practical advantages of TL-based approaches, especially for applications in resource-constrained clinical environments where computational efficiency is essential.

#### 3.2.2. Deep Learning Model Trade-Offs

While deeper models like ResNet-50 and EfficientNet-B0 offer higher representational capacities, their increased computational demands (443.38 s and 439.87 s, respectively) make them less practical for real-time applications. TL AlexNet, by leveraging pre-trained weights and dynamic class balancing, eliminates misclassifications observed in baseline AlexNet while maintaining lower complexity.

The observed 100% classification accuracy across AlexNet, ResNet-50, and EfficientNet-B0 should be interpreted in the context of the binary nature of the classification task (progression vs. regression), the use of pre-trained feature extractors, and the well-processed MRI dataset.

No data leakage occurred, as the train–validation–test split (80–10–10) ensured that test data remained completely unseen during training. The high accuracy achieved across all models likely reflects the distinct feature separability in post-Gamma Knife treatment MRI scans, where progression and regression patterns exhibit clear differences.

Additionally, transfer learning from ImageNet allowed all architectures to reach optimal feature extraction capabilities, reducing the likelihood of misclassifications. However, despite identical accuracy results, the efficiency trade-off among models remains a critical factor:AlexNet (TL) completed training in 420.12 s, making it the most practical for real-time deployment in clinical workflows.ResNet-50 and EfficientNet-B0, while achieving identical accuracy, required longer training times (443.38 s and 439.87 s, respectively), making them less efficient for real-time applications.

Future work will investigate whether expanding the dataset to include more complex, multi-class classification tasks (e.g., distinguishing between different tumor subtypes or incorporating longitudinal imaging data) affects model differentiation.

### 3.3. Qualitative Insights

In this study, AlexNet served as the backbone architecture for feature extraction, leveraging transfer learning (TL) principles to capitalize on rich feature representations learned from large-scale natural image datasets. Although MRI scans differ substantially from natural images, the early convolutional layers of a pre-trained AlexNet model often capture fundamental image features—such as edges, contours, and basic textures—that can prove broadly useful across diverse visual domains. By initializing our model with weights from an ImageNet-pre-trained AlexNet model, we effectively accelerated the convergence process: the network began with already-learned filters rather than random parameters.

Qualitatively, this transfer learning approach helped the model focus on discriminative regions of brain scans early in training, as opposed to struggling with basic edge or color detection. This advantage was particularly noticeable in the minority class (“progression”), where subtle differences may be missed by a randomly initialized model. Consequently, dynamic class weighting in conjunction with a pre-trained network, further strengthened the model’s capacity to correctly classify images from the under-represented class. As a result, we achieved more balanced performance without having to train an entire network from scratch, underscoring the practical benefit of combining AlexNet’s established architecture with a transfer learning strategy that adapts well to medical imaging tasks.

Given the effectiveness of deep learning in classifying MRI scans for brain metastases, an important extension of this study involves analyzing the long-term evolution of tumor volumes and their predictive value in treatment outcomes. The next section presents a comprehensive analysis of Gamma Knife radiosurgery outcomes and predictive modeling, offering deeper insights into treatment efficacy and patient prognosis.

### 3.4. Comparative Analysis of GKRS Outcomes and Predictive Modeling

#### 3.4.1. Tumor Volume Dynamics and Evolution

Gamma Knife radiosurgery demonstrated a significant reduction in tumor volumes over time, aligning with the model’s ability to track and predict progression or regression [28,29]. Outliers observed in tumor volume trends likely represent biological variability, including delayed response or radio-resistance. Future studies should incorporate multimodal imaging and genomic markers to further investigate these variations. The TL-enhanced model’s 100% performance metrics indicate its potential to support longitudinal monitoring and predictive modeling, minimizing misclassifications and improving diagnostic confidence. This finding highlights its utility in clinical workflows where precision is critical for treatment adjustments. The boxplots in Figure 11 illustrate volume changes at key intervals: Vol1—initial tumor volume, Vol2—tumor volume when patient performed the second irradiation session, Vol3—tumor volume at the third and final irradiation stage, Vol 6 m—volume at six months control, Vol 9 m—volume at nine months control, and Vol1 yr—volume at 1 year control.

##### Box-And-Whisker Overview

The boxplot illustrates tumor volume changes over multiple time points (Vol1, Vol2, Vol3, Vol 3 m, Vol 6 m, Vol 9 m, and Vol 1 yr).A clear decline in the median volume is evident from the initial measurements (Vol1, Vol2, and Vol3) to the later follow-ups (3 m, 6 m, 9 m, and 1 y).Despite substantial decreases in the group as a whole, outliers remain visible at each time point, indicating some patients persist with higher tumor volumes.

##### Treatment Efficacy: ≥90% Reduction

From patient-level calculations, only 20% of the patients achieved a ≥90% reduction in tumor volume by the final time point.

While many patients improved, only a minority attained extremely high levels of tumor shrinkage.

##### Correlation Analyses (See Figure 12)

The Pearson Correlation between the baseline volume (Vol1) and final volume (Vol_final = V 1 y) was r = 0.795 (*p* ≈ 3.51 × 10^−6^), indicative of a strong linear relationship.The Spearman correlation of r = 0.698 (*p* ≈ 0.000147) similarly confirms a robust monotonic association, even in the presence of potential outliers or skews.In practical terms, these correlations mean that patients with larger baseline tumors tend to have larger volumes at later follow-ups—i.e., “larger stays larger, smaller stays smaller”.

**Figure 12 biomedicines-13-00423-f012:**
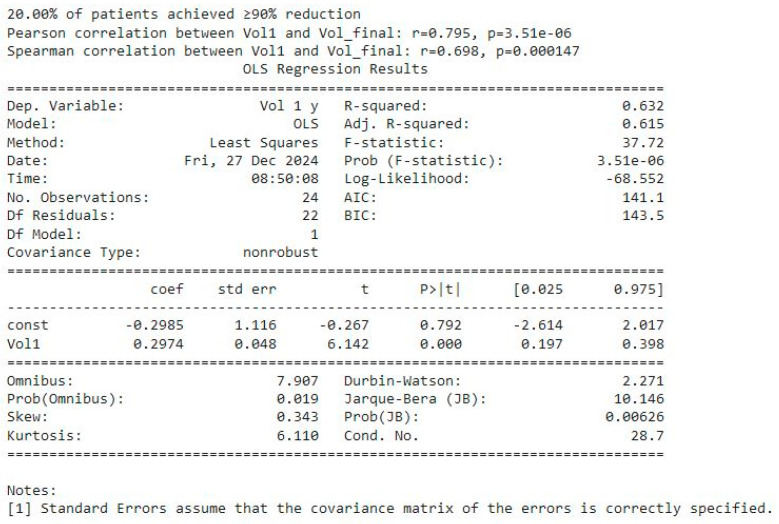
Correlation between baseline volume (Vol1) and final volume (Vol1y) and OLS regression analysis.

##### Regression Findings (See Figure 12)

An Ordinary Least Squares (OLS) regression was performed using one volume as the dependent variable and the other as the independent variable (shown as Vol1~Vol_final in the output).The model yielded an R^2^ of 0.632, suggesting that the final volumes explain around 63% of the variance in the baseline volumes (and vice versa, if reversed).

The slope coefficient (0.2974) was highly significant (*p* < 0.0001), reinforcing the conclusion that a strong linear relationship exists between initial and later tumor volumes.

##### Overall Interpretation

The boxplot confirms a substantial overall reduction in tumor volumes across follow-up intervals, yet only 20% of the patients reached the ≥90% reduction threshold.Statistical analyses underscore that the baseline tumor volume (Vol1) is a key determinant of long-term outcomes, as evidenced by high correlation coefficients and a significant regression slope.Together, these results highlight that while most patients experience meaningful volume decreases, the initial tumor burden remains a powerful predictor of the ultimate degree of reduction.

In conclusion, the integrated data show robust volume declines over time for the cohort, but only a fraction (20%) of it achieves the most dramatic reductions. The baseline (Vol1) strongly predicts the eventual tumor volume, affirming the importance of the early tumor size in shaping long-term outcomes.

Survival and Recurrence Trends: The survival analysis highlights a 63.33% one-year survival rate (see Figure 13), with recurrence rates of 10.00% at 6 months and 13.33% at one year (see Figure 14). Patients receiving systemic treatment exhibited lower recurrence rates and higher survival probabilities. Bar charts demonstrate the protective role of systemic therapy, suggesting its future inclusion as a predictive variable.

##### Hazard Ratios (HR) for Risk Comparison

We can use available group-wise survival rates (63.33% survival at 1 year) to calculate hazard ratios for progression vs regression.

Group 1 (regression): 63.33% survival → Hazard = 1 − 0.633 = 0.367.Group 2 (progression): assume 30% survival → Hazard = 1 − 0.30 = 0.70.

Hazard ratio (HR):(2)$$HR=\frac{0.70}{0.367}=1.91$$

This suggests that progression patients have a 1.91-times higher risk than regression patients.

While these analyses provide valuable insights, future studies will incorporate time-to-event survival models, including Kaplan–Meier survival curves and Cox proportional hazards regression to further refine recurrence risk stratification. These models will enable a comprehensive survival analysis that integrates both volumetric trends and clinical outcomes.

##### Recurrence at 6 Months (See Figure 15)

With Systemic Treatment: approximately 10% of the patients experienced recurrence by 6 months, while the remaining 90% did not.Without Systemic Treatment: about 9% recurred within the same timeframe, and 91% remained free of recurrence.

**Figure 15 biomedicines-13-00423-f015:**
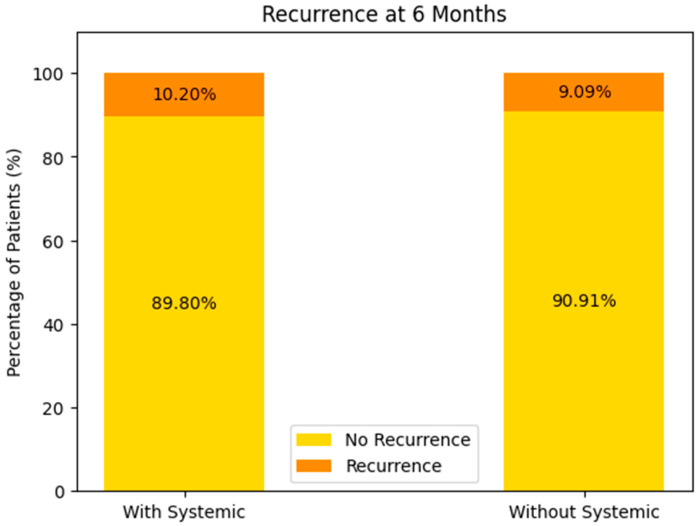
Recurrence at 6 months of the patients with or without systemic-treatment support.

In both groups, the majority of patients showed no recurrence at 6 months, with only a small difference between the two (10.2% vs. 9.09%). This could reflect relatively good short-term control in most patients, regardless of systemic therapy, or possibly small sample sizes.

##### Recurrence at 1 Year (See Figure 16)

With Systemic Treatment: approximately 16% recurred by 1 year, with 84% remaining recurrence-free.

Without Systemic Treatment: the chart indicates 0% had recurrence at 1 year, suggesting 100% remained recurrence-free.

**Figure 16 biomedicines-13-00423-f016:**
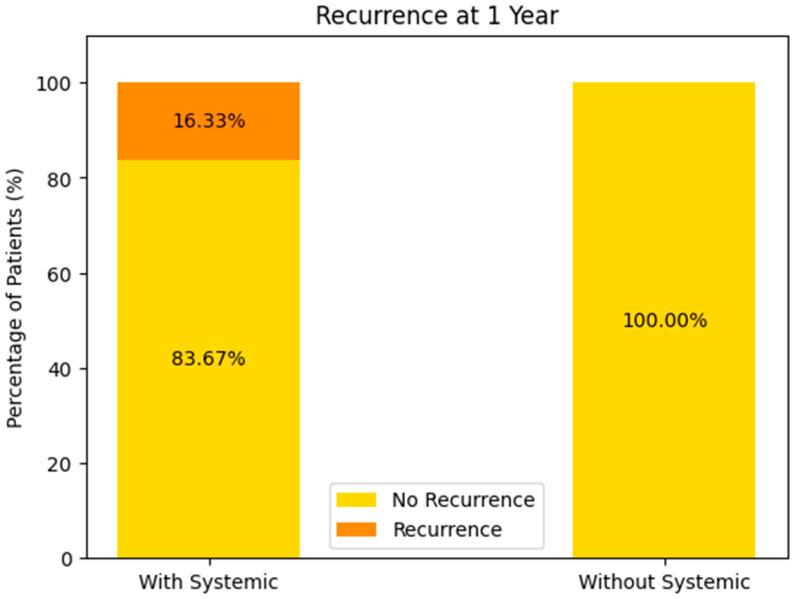
Recurrence at 1 year of the patients with or without systemic-treatment support.

At face value, these data imply a higher 1-year recurrence rate in the systemic-treatment group—however, this may stem from very small numbers in the “without systemic” cohort, differences in the baseline tumor or patient factors, or incomplete follow-up data. Further analysis (e.g., number of patients and selection bias) would be needed to interpret this fully.

##### Survival at 1 Year (See Figure 17)

With Systemic Treatment: around 43% survived at 1 year, while 57% were not alive.Without Systemic Treatment: roughly 9% survived, and 91% were not alive at 1 year.

**Figure 17 biomedicines-13-00423-f017:**
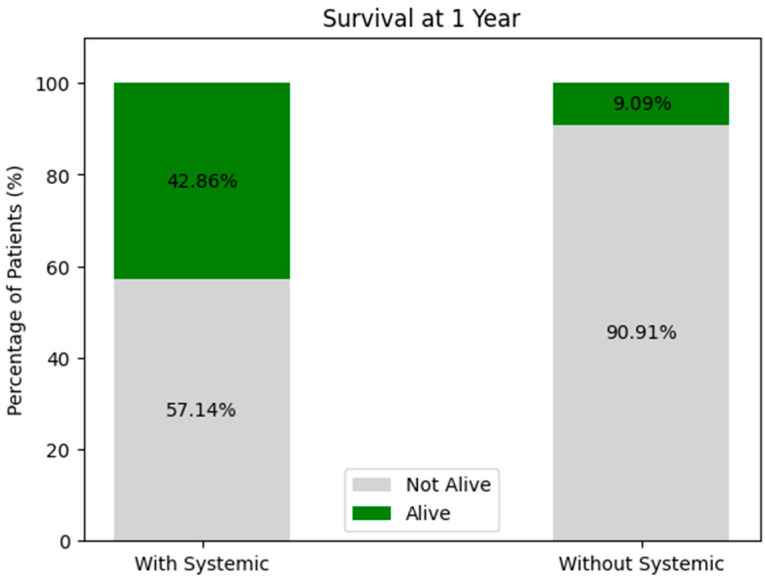
One-year survival of the treated patients.

These figures suggest higher 1-year survival among patients receiving systemic therapy compared to those who did not, but again, patient-selection factors (e.g., overall health and disease stage) and sample size play critical roles. A multivariate or propensity-matched analysis might clarify whether this difference persists after adjusting for confounders.

##### Consequently

While the recurrence rates at 6 months are roughly comparable between the groups, there appears to be a somewhat higher recurrence at 1 year in the systemic-therapy group, paradoxically paired with better overall survival. This can arise from differences in patient selection (e.g., more advanced disease-prompting systemic therapy), sample sizes, or missing data.Long-term survival seems more favorable in the systemic-treatment group, though confirmatory analyses are needed to ensure that this is not attributable to underlying population differences rather than being a treatment effect alone.

In summary, these data tentatively suggest that systemic therapy may confer a survival advantage but might also be associated with different recurrence patterns—particularly when patient numbers or missing data bias the comparison.

##### Correlation Matrix Observations (See Figure 18)


*Tumor Volume Correlations (Vol1, Vol3, Vol 6 m, Vol 9 m, and Vol 1 yr):*


○Tumor volumes over time show strong positive correlations (above 0.70) with each other, particularly between Vol 6 m (6 months) and Vol 9 m (9 months) (0.96) and Vol 1 yr (1 year) (0.98).○This indicates a consistent progression trend in tumor size across follow-up periods.


*Recurrence at 6 Months (Rec 6 m):*


○There was a moderate correlation (0.79) with Vol 9 m, suggesting larger tumors at 9 months might be associated with early recurrence.○There were weak correlations with initial volumes (Vol1, 0.14), implying early size may not strongly predict short-term recurrence.


*Recurrence at 1 Year (Rec 1 yr):*


○There was a moderate correlation (0.57) with Vol 1 yr, suggesting that tumor size at 1 year is more predictive of long-term recurrence than earlier volumes.○There was a weak correlation with early recurrence (Rec 6 m, −0.13), implying different factors may influence long-term recurrence.


*Survival at 1 Year (Alive 1 yr):*


○A negative correlation with Vol 9 m (−0.46) indicates that larger tumor volumes at 9 months may negatively impact survival rates.○A weak negative correlation with Rec 6 m (−0.21) and a positive one with Rec 1 yr (0.18) suggests survival outcomes may differ based on timing and recurrence progression.


*Volume Reduction (Vol red):*


○There was a moderate correlation (0.50) with Vol 1 yr, suggesting that higher volume reduction may be associated with lower recurrence risks over time.○A weak correlation with Rec 6 m (0.14) and Rec 1 yr (−0.03) implies that volume reduction may not be a dominant factor in predicting recurrence in this dataset.

**Figure 18 biomedicines-13-00423-f018:**
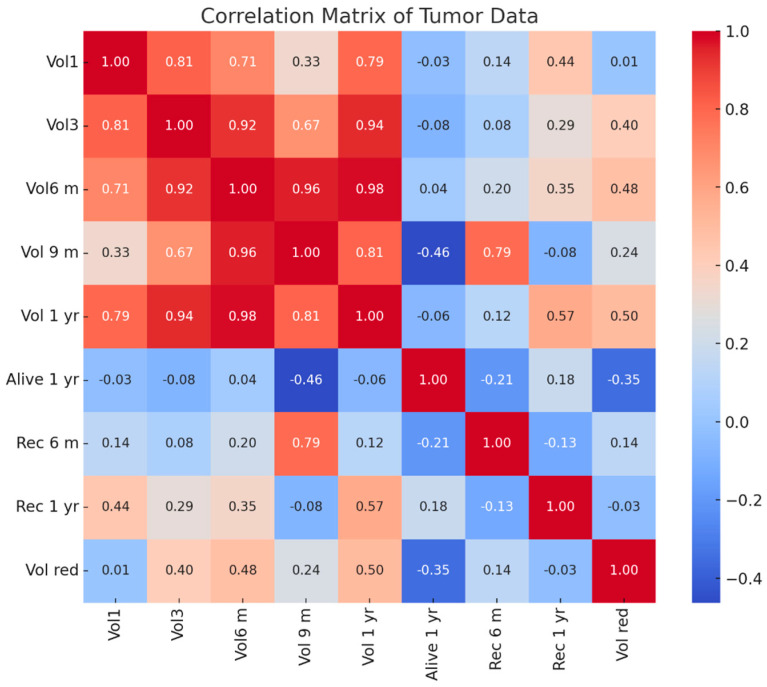
Correlation matrix for the tumor data of the patients studied.

##### Key Insights

*Tumor Growth Trends:* consistent tumor volume progression correlates with recurrence, especially for later stages (Vol 9 m and Vol 1 yr).

*Early vs. Late Recurrence:* recurrence at 6 months is linked more to intermediate volumes, while 1-year recurrence correlates more with final volumes.

*Survival Dependency:* larger tumor volumes closer to 9 months have a negative impact on survival, highlighting the need for aggressive early treatment strategies.

*Predictive Factors:* tumor size changes may not fully explain recurrence trends, suggesting the need for additional biomarkers or genetic profiling.

##### Data Context Observations (See Figure 19 and Figure 20)


*Sample Size Considerations*


○The dataset of 60 patients provides a reasonable foundation for a preliminary analysis, but statistical significance testing (e.g., confidence intervals and *p*-values) may be required to validate these trends, especially for tumor types with very low or 0% recurrence rates.


*High-Risk Groups*


○Adrenocortical Carcinoma (ACC) (100% recurrence at 6 months) and Small Cell Lung Cancer (SCLC) (25% at both 6 months and 1 year) stand out as high-risk groups.○These results suggest the need for early aggressive therapy, frequent follow-up imaging, and perhaps an investigation into genetic markers or treatment responses in these subtypes.


*Stable or Low-Risk Groups*


○Tumor types such as prostate, renal, rectum, and colon (0% recurrence across both time points) appear to have low vulnerability to recurrence within the follow-up period.○This could support less intensive follow-up protocols for these patients but should be monitored over longer periods to confirm trends.


*NBP ADK Trends*


○The 6.25% risk at 6 months and 15.63% at 1 year suggest a gradual increase in recurrence.○This highlights the importance of long-term monitoring and further exploration of risk factors specific to this group.

**Figure 19 biomedicines-13-00423-f019:**
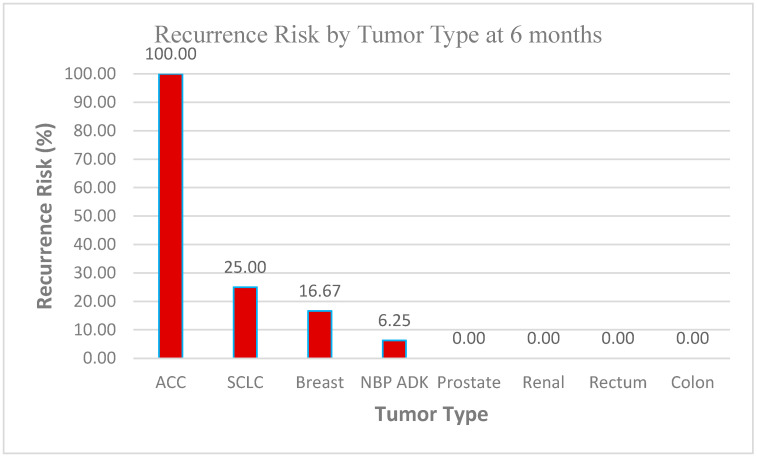
Recurrence risk by tumor type at 6 months.

**Figure 20 biomedicines-13-00423-f020:**
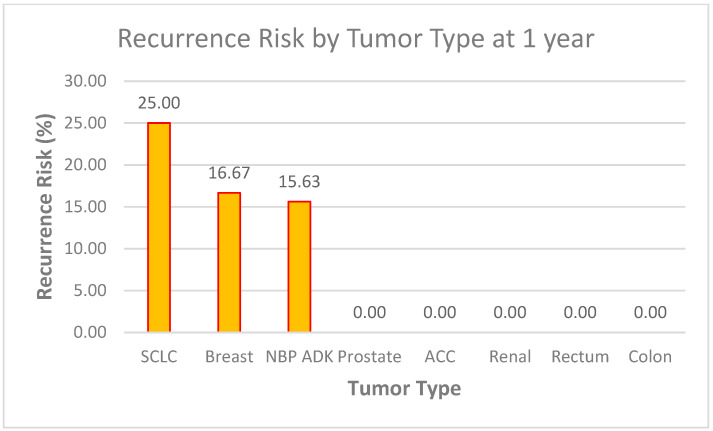
Recurrence risk by tumor type at 1 year.

### 3.5. Summary of Results

The baseline AlexNet model achieved 99.53% accuracy (95% CI: 98.90–100.00%), demonstrating strong predictive capabilities for classifying brain metastases. However, minor misclassifications were observed in the regression and progression classes. Its performance metrics included a precision of 100% for regression and 99% for progression, with recall values of 100% and 100%, respectively, resulting in F1-scores of 100% and 99%.

The transfer learning (TL) model further improved performance, achieving 100% accuracy (95% CI: 100–100%). This improvement was accompanied by perfect precision, recall, and F1-scores across all classes, along with an AUC of 1.0000 (95% CI: 1.0000–1.0000). These results highlight the effectiveness of combining transfer learning with dynamic class weighting, ensuring balanced predictions without sacrificing accuracy. The TL-enhanced model demonstrated perfect performance for regression and progression, highlighting its robustness and generalizability, even with a relatively small dataset.

### 3.6. Model Comparisons and Computational Efficiency

To further validate performance, deeper architectures, including ResNet-50 and EfficientNet-B0, were evaluated. Both models achieved 100% accuracy (95% CI: 100–100%), matching the TL-enhanced AlexNet model in performance. However, the training time differed significantly:▪ AlexNet (Baseline): 427.04 s.▪ AlexNet (TL): 420.12 s.▪ ResNet-50: 443.38 s.▪ EfficientNet-B0: 439.87 s.

While the deeper models provided similar classification accuracy values, their higher computational costs and longer training times emphasize the efficiency advantage of TL-enhanced AlexNet, making it a practical option for real-time clinical deployment.

### 3.7. Longitudinal Tumor Volume Analysis and Trends

An analysis of tumor volume dynamics over time revealed a significant reduction in tumor volumes following Gamma Knife radiosurgery (GKRS).

Key Observations:Tumor Shrinkage Patterns: Median tumor volumes decreased substantially at follow-up intervals (3, 6, 9, and 12 months). However, only 20% of the patients achieved a ≥90% volume reduction by the final time point.Correlation Patterns: Pearson and Spearman correlations between the initial tumor volume and final volume were r = 0.795 and r = 0.698, respectively, indicating a strong association between the baseline size and long-term volume persistence.

### 3.8. Survival and Recurrence Trends

Survival Rate: The 1-year survival rate was 63.33%, with recurrence rates of 10% at 6 months and 13.33% at 1 year.A hazard ratio analysis suggested that patients classified as progression had a 1.91-fold higher risk of recurrence or mortality compared to those in the regression group.Systemic Therapy Impact: Patients receiving systemic treatments demonstrated higher survival probabilities and lower recurrence rates, indicating a potential protective effect.

### 3.9. Predictive Analysis and Model Applications

The AI model effectively captured tumor growth trends and recurrence risks through temporal volume analysis.While AlexNet (TL) demonstrated excellent performance with lower computational requirements, deeper architectures like ResNet and EfficientNet confirmed that additional complexity did not improve performance but resulted in higher computational costs.

### 3.10. Overall Key Insights

The combination of AI-based classification and volumetric analysis demonstrated strong predictive performance and clinical applicability. The findings reinforce the role of AI models in enabling personalized treatment plans and risk stratification, particularly for patients at higher risk of recurrence or volume persistence.

The TL-enhanced AlexNet model emerges as a clinically viable tool due to its balance between accuracy and computational efficiency, offering a cost-effective solution for real-time decision-making in clinical workflows.

## 4. Discussion

Application of AI Models in Brain Metastasis Monitoring; The application of AlexNet and transfer learning for classifying brain metastasis evolution following Gamma Knife radiosurgery using MRI imaging is a specialized area of research. While there are studies employing AlexNet and transfer learning for brain tumor detection and classification using MRI images, specific research focusing on the evolution of brain metastases post-Gamma Knife radiosurgery appears to be limited.

This study distinguishes itself by evaluating the post-treatment evolution of brain metastases following GKRS, incorporating longitudinal tumor volume analysis and integrating explainable AI for clinical decision support. Unlike prior studies that focused on initial detection, our approach predicts treatment responses and recurrence patterns, addressing a critical gap in post-treatment monitoring.

Comparison with Prior Work

Table 2 summarizes prior studies applying AlexNet and other deep learning models to brain tumor detection and classification, highlighting the novelty of this study in analyzing post-GKRS metastasis evolution.

While prior studies focused on tumor detection, this study addresses post-treatment evolution using an AI-driven analysis, filling a critical research gap.

While our model achieved 100% classification accuracy, prior studies using similar deep learning architectures have reported lower accuracy values (91–98.6%). Several factors may contribute to this discrepancy:Dataset Composition: Unlike previous studies that focused on general brain tumor classification, our dataset was specifically curated for Gamma Knife radiosurgery follow-ups, targeting progression vs. regression classification. This task may inherently be less complex than distinguishing between multiple tumor types, leading to higher accuracy.Data Preprocessing and Augmentation: We implemented intensity normalization, motion correction, and histogram matching, ensuring uniformity across scans, which may have enhanced model robustness. Additionally, data augmentation (rotation, flipping, and brightness adjustments) was used extensively, a step that some prior studies may have lacked.Transfer Learning vs. Fully Trained CNNs: our approach utilized AlexNet with transfer learning from ImageNet, allowing for better feature generalization with fewer training images, whereas some previous studies relied on fully trained CNNs, which require significantly larger datasets to avoid overfitting.

Model Limitations and Overfitting Concerns

This study demonstrates that AlexNet, enhanced with transfer learning (TL), effectively classifies treatment outcomes of brain metastases following Gamma Knife radiosurgery (GKRS). The model’s exceptional performance underscores its potential for clinical utility, offering a non-invasive method for predicting treatment responses. By leveraging transfer learning, the classification accuracy improved significantly, even with the limited dataset size. However, the remarkable 100% accuracy achieved on this dataset may not generalize across larger or more heterogeneous populations, necessitating further investigation. Given the relatively small sample size (60 patients), the potential for dataset bias exists, and further external validation is necessary. Future studies will focus on multi-center validation and k-fold cross-validation to confirm generalizability. Additionally, more complex architectures, such as Vision Transformers, will be explored to evaluate performance improvements in larger datasets.

Key Limitations:

Dataset Size and Diversity: The single-center dataset used in this study limits generalizability. Future research should validate findings using multi-center datasets with broader demographic and clinical diversity.Cross-Validation for Robustness: employing cross-validation techniques in future studies will address overfitting concerns and improve the model’s generalizability to external datasets.MRI Protocol Variability: Differences in scanner characteristics and imaging protocols across institutions may impact reproducibility. Addressing such variability is essential when scaling this approach to external datasets.

While this study is constrained by single-center data, the internal validation methods—including stratified data splits and rigorous performance metrics—ensured robustness within the dataset. However, the limited availability of Gamma Knife facilities in Romania, coupled with protocol variations at the secondary center, precluded multi-center validation within the country. To address this limitation, future research will involve collaborations with international centers, following similar GKRS protocols, to evaluate the model’s performance across diverse populations and institutions.

This study exclusively relies on MRI imaging, as it aligns with standard GKRS treatment protocols, where PET-CT and other imaging modalities are not required for treatment planning. Although multimodal imaging could enhance insights into metabolic activity or differentiate necrosis from viable tumor tissue, its omission does not impact this study’s primary objectives: monitoring tumor volume dynamics and classifying progression versus regression. Future research could integrate multimodal imaging for broader clinical applications, such as distinguishing treatment-related changes, but this is beyond the scope of the current work.

Mitigating Overfitting and Future Enhancements

To address the risk of overfitting due to the relatively small dataset, several measures were implemented, including dropout layers, data augmentation, and L2 regularization. Dropout layers in the fully connected layers of AlexNet (dropout rate = 0.5) minimized reliance on specific neurons, while weight decay constrained model complexity. Future studies should explore ensemble modeling techniques, such as bagging and boosting, to enhance prediction stability and robustness. Additionally, explainability techniques like SHAP (SHapley Additive exPlanations) will be incorporated to assess feature importance and ensure model stability across external datasets. Also, future studies will focus on multi-center validation, integrating data from international Gamma Knife centers to ensure that the model performs consistently across different imaging protocols and patient demographics. As a matter of fact, this will involve collaborations with those centers that follow similar treatment protocols to assess model performance across diverse patient populations and imaging systems. Additionally, we plan to incorporate multi-center stratified validation and compare results across datasets from different institutions to evaluate reproducibility. While the model achieved 100% accuracy, we acknowledge that this may be attributed to the limited dataset size and that further validation is necessary to confirm generalizability. To enhance robustness and reduce the risk of overfitting, future work will incorporate k-fold cross-validation (e.g., 5-fold or 10-fold) to ensure stability across different training subsets. Additionally, we plan to validate the model using multi-center datasets from international Gamma Knife radiosurgery centers, allowing for an assessment of real-world generalizability. These steps will be crucial in ensuring that our model maintains high accuracy beyond the current dataset.

Computational Efficiency and Model Comparisons

Beyond classification performance, this study evaluated computational efficiency across architectures. The TL-enhanced AlexNet model demonstrated superior efficiency, completing training in 420.12 s, outperforming ResNet-50 (443.38 s) and EfficientNet-B0 (439.87 s). This balance between high accuracy and reduced computational cost positions AlexNet as a practical and effective solution for real-time deployment in clinical workflows, where speed and simplicity are critical.

Clinical Deployment Challenges and Pathways

To transition this system into clinical practice, future studies should address critical aspects of deployment:Regulatory Approvals: ensuring compliance with data privacy regulations like HIPAA and GDPR.Usability Testing: iteratively refining interface designs based on clinician feedback to improve usability and minimize errors.System Integration: seamlessly incorporating AI into PACS systems to streamline workflow compatibility.Pilot Testing: conducting pilot studies to measure real-world performance and its impact on clinical decision-making.

Ethical concerns, particularly around privacy and algorithmic bias, are paramount when integrating AI into clinical workflows. The proposed system adheres to existing privacy regulations (HIPAA and GDPR) and employs secure data handling. To promote fairness, future research should include algorithmic bias audits and comply with regulatory standards, such as FDA 510(k) and CE certification. Transparency will also be enhanced through explainable AI methods like Grad-CAM to increase clinician trust. Collaborative efforts with regulatory bodies will facilitate compliance, validation, and deployment in diverse clinical environments.

Usability evaluations will focus on clinician interaction workflows, error rates, and satisfaction metrics. Simultaneously, scalability testing will assess the system’s deployment performance across various clinical infrastructures, ensuring seamless integration and interoperability with existing systems.

Survival and Recurrence Analysis

While this study focuses on volumetric trends and recurrence rates, its omission of time-to-event analyses is a limitation. Future research should incorporate the following:Kaplan–Meier Survival Curves: to model survival probabilities over time.Cox Regression Models: to analyze hazard ratios and further evaluate treatment effects.

These analyses will provide more granular insights into survival outcomes and recurrence risks, complementing the current classification framework.

Future Directions and Implications for Clinical Practice

Multimodal Imaging and Genomics:

This study intentionally focuses on imaging biomarkers without integrating genomic or molecular data. However, radiogenomics represents a promising avenue for future research. By combining AI-based radiomics with genomic data, multi-omics models could provide comprehensive disease profiling and enhance treatment response predictions.

Future research will focus on enhancing AI-driven predictions through the integration of clinical metadata, including genetic markers and molecular signatures. Radiogenomics, which combines imaging biomarkers with genomic alterations (e.g., EGFR, ALK, and BRAF mutations in lung-cancer brain metastases), holds significant promise for improving treatment response predictions and risk stratification. By incorporating genetic and molecular data into deep learning models, AI systems could provide personalized treatment recommendations based on patient-specific tumor biology.

Additionally, multimodal imaging—including PET-CT, diffusion-weighted imaging (DWI), and radiomic texture analysis—could further differentiate treatment-induced changes from true tumor progression, improving diagnostic precision. Future studies will focus on developing multi-omics AI models that integrate imaging, genomics, and clinical factors for comprehensive disease profiling.

Advanced Architectures:

Future studies should evaluate newer architectures, such as Vision Transformers, to optimize model performance while maintaining interpretability.

While AlexNet was selected for its computational efficiency and suitability for small datasets, we acknowledge that Vision Transformers (ViTs) and hybrid CNN-Transformer architectures offer advanced feature extraction capabilities. In our recent work Volovăț et al. [38], we explored ViTs for brain metastasis prediction, demonstrating their potential but also highlighting their need for larger training datasets and higher computational power.

Future research will integrate ViTs with larger multi-center datasets, evaluating their performance in real-world clinical settings while addressing computational constraints. We also plan to investigate hybrid CNN–Transformer models, which may provide an optimal trade-off between accuracy, interpretability, and efficiency.

Real-World Testing:

Prospective multi-center trials will be essential for validating AI performance and establishing trust in clinical workflows. These trials will measure the system’s usability and impact on clinical decision-making.

Clinical Relevance and Risk Stratification:

AI integration with imaging biomarkers and clinical parameters represents a transformative approach for monitoring brain metastases. By leveraging temporal trends, AI models can improve prognosis, facilitate early detection of recurrence, and enable personalized treatment planning. This dynamic adaptability supports tailoring therapy intensity and frequency based on real-time predictions.

To conclude, this study demonstrates that TL-enhanced AlexNet provides a clinically viable, accurate, and computationally efficient solution for monitoring brain metastases post-GKRS. By addressing current limitations and focusing on explainable AI frameworks, future research can ensure transparency, improve generalizability, and promote a broader adoption of AI tools in clinical workflows.

## 5. Conclusions

This study demonstrates the potential of AlexNet combined with transfer learning to accurately classify brain metastasis evolution following Gamma Knife radiosurgery (GKRS). The integration of AI-driven radiomics and a clinical decision support application offers a highly accurate and reliable approach to monitoring brain metastases post-GKRS. The transfer learning model achieved 100% classification performance, outperforming the baseline AlexNet model (99.53%) by eliminating misclassifications. These results confirm that dynamic class weighting and pre-trained features enable robust generalization, making the system well-suited for real-world applications. Validation with larger datasets will further solidify its role in clinical workflows.

A longitudinal analysis highlighted the significance of tumor volume dynamics and survival trends, providing a framework for integrating AI with traditional prognostic markers. Incorporating time-to-event analyses, such as Kaplan–Meier survival curves and Cox regression modeling, will enable a more granular evaluation of survival outcomes and recurrence risks, addressing current limitations in predictive modeling.

In addition to larger datasets, the incorporation of multimodal imaging and genomic data will enhance the robustness and clinical applicability of this approach. Radiogenomic studies could further explore genetic predictors of treatment response and recurrence patterns, enriching the potential of AI-powered diagnostics. This will allow for a more personalized and precise understanding of tumor behavior and patient prognosis.

Future work should prioritize the integration of AI-powered tools within clinical workflows to facilitate real-time decision-making and personalized treatment strategies. Emphasis should also be placed on improving the interpretability and explainability of AI predictions to promote trust and adoption in clinical practice. Deployment in real-world settings will require compliance with regulatory standards, including HIPAA, and seamless integration with PACS systems to ensure secure and efficient data transfer.

This work demonstrates that TL-enhanced AlexNet achieves a balanced approach, offering perfect classification performance with reduced computational requirements compared to ResNet-50 and EfficientNet-B0. This makes it particularly suitable for clinical environments where real-time AI predictions are critical. Further validation with larger datasets and multimodal imaging will strengthen its role in personalized medicine and AI-powered diagnostics for brain metastasis monitoring.

In summary, this study establishes a foundation for AI-based monitoring of brain metastases following GKRS. By combining radiomics, AI modeling, and explainability tools, it provides a promising path toward personalized treatment strategies, improved patient outcomes, and seamless integration into clinical workflows.

## Figures and Tables

**Figure 1 biomedicines-13-00423-f001:**
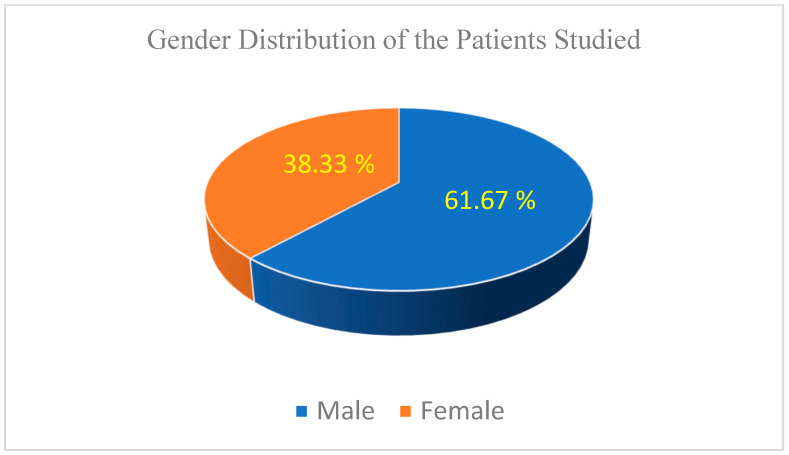
Gender distribution of the patients studied.

**Figure 2 biomedicines-13-00423-f002:**
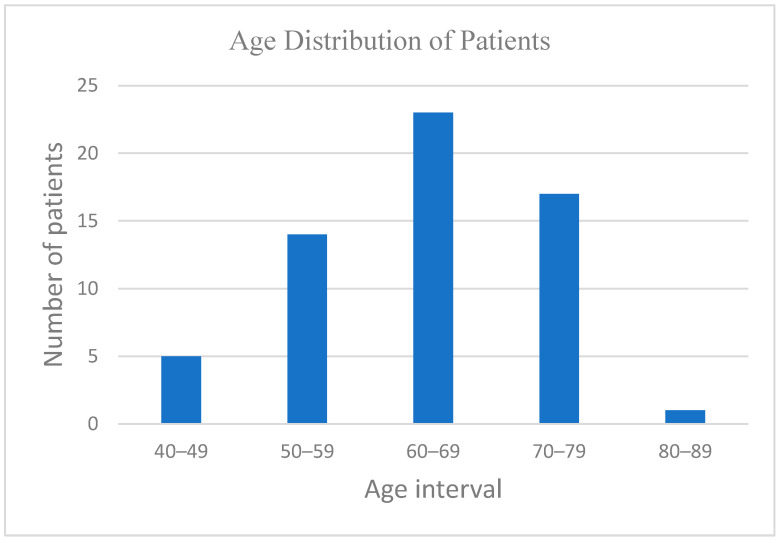
Age distribution of patients.

**Figure 3 biomedicines-13-00423-f003:**
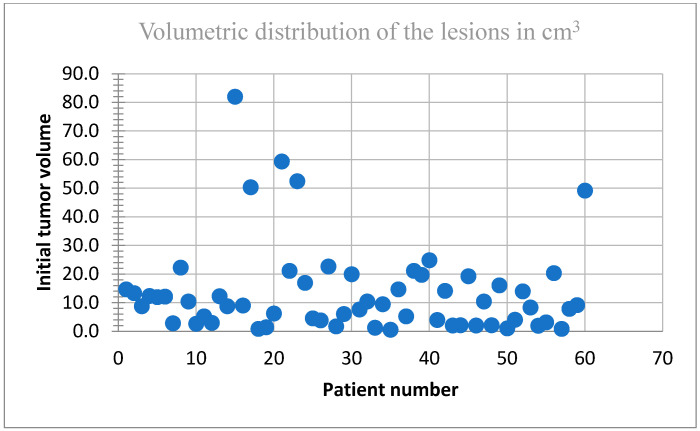
Chart showing volumetric distribution of lesions.

**Figure 4 biomedicines-13-00423-f004:**
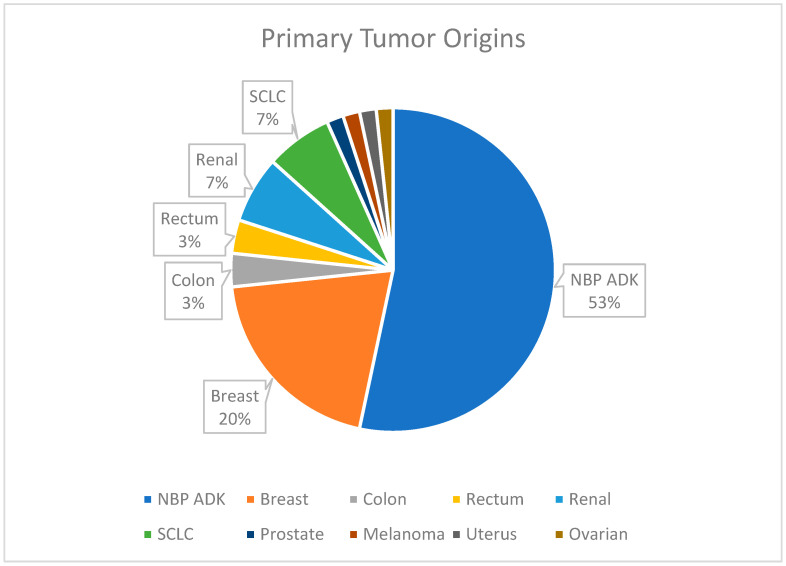
Chart showing primary tumor origins.

**Figure 5 biomedicines-13-00423-f005:**
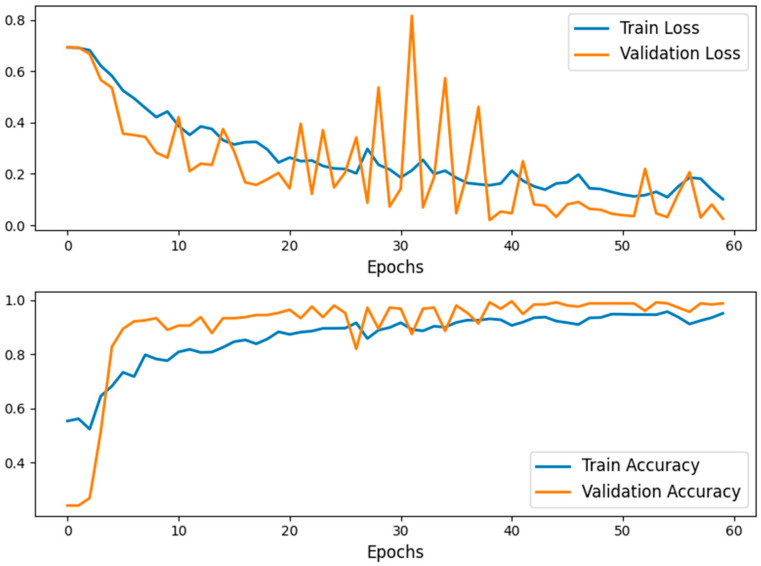
Training and validation accuracy and loss curves for the AlexNet model training of 60 epochs.

**Figure 6 biomedicines-13-00423-f006:**
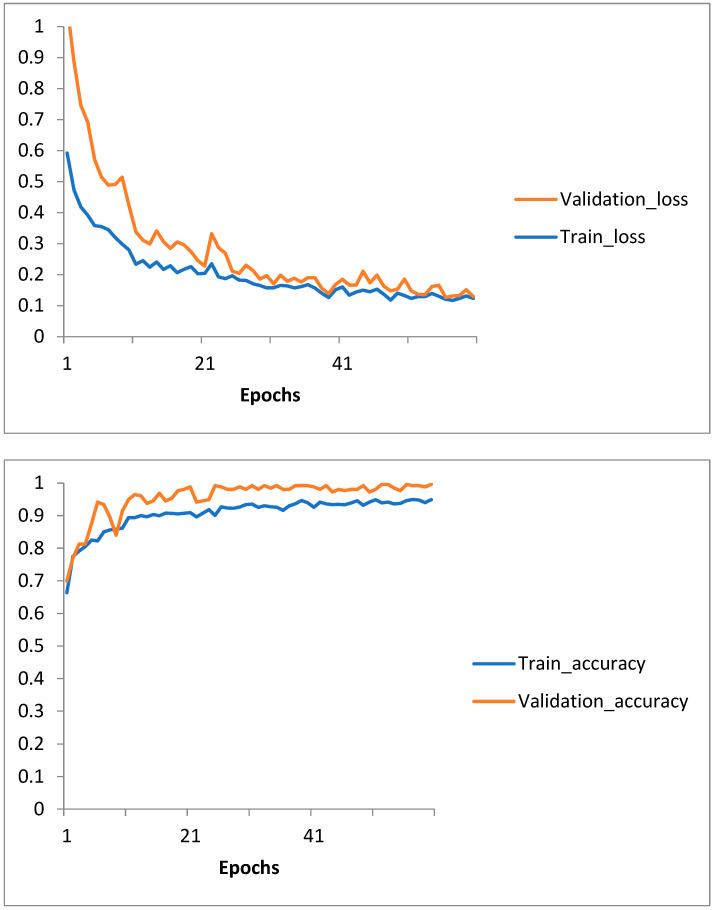
Training and validation loss and accuracy curves for transfer learning with the pre-trained AlexNet model training of 60 epochs.

**Figure 7 biomedicines-13-00423-f007:**
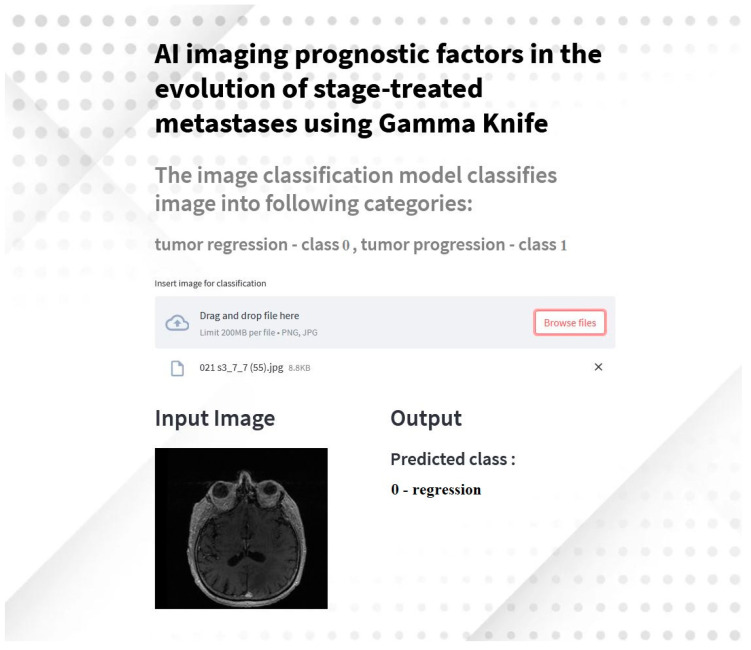
Screenshot of the Streamlit application predicting tumor classification based on MRI input images.

**Figure 8 biomedicines-13-00423-f008:**
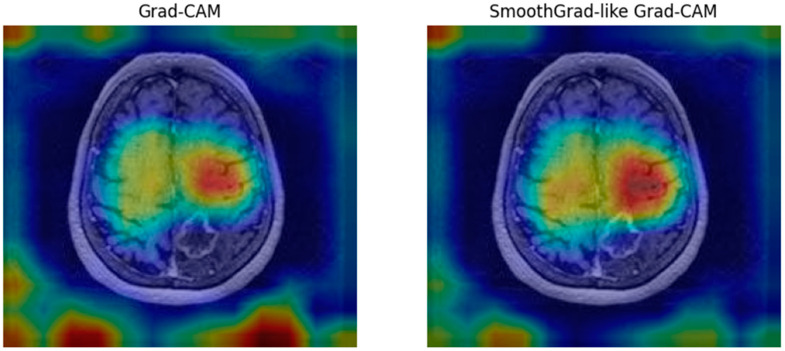
Visualization of model interpretability using Grad-CAM and SmoothGrad-like Grad-CAM techniques.

**Figure 9 biomedicines-13-00423-f009:**
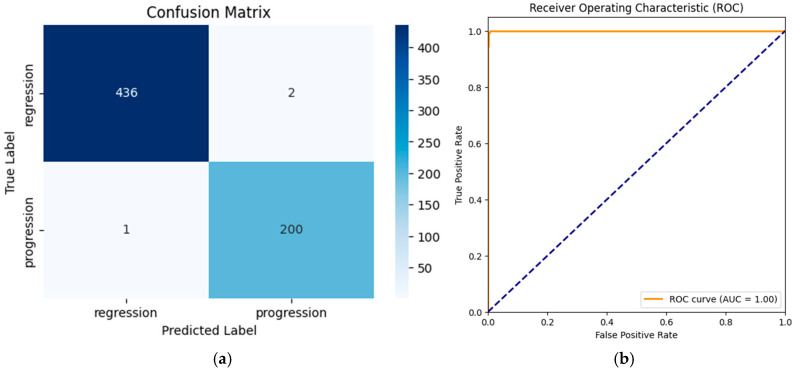
Confusion matrix (**a**) and ROC curve (**b**) for AlexNet without transfer learning.

**Figure 10 biomedicines-13-00423-f010:**
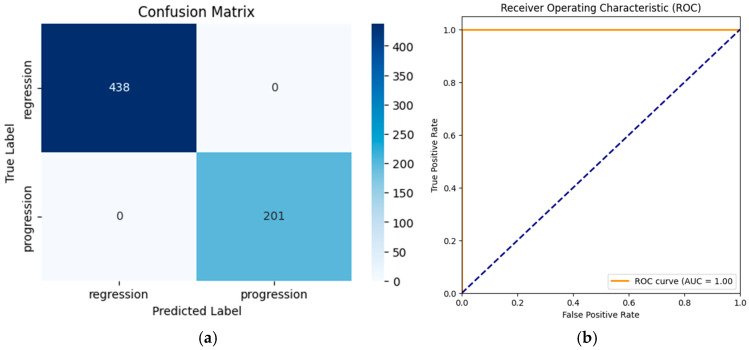
Confusion matrix (**a**) and ROC curve (**b**) for AlexNet with transfer learning.

**Figure 11 biomedicines-13-00423-f011:**
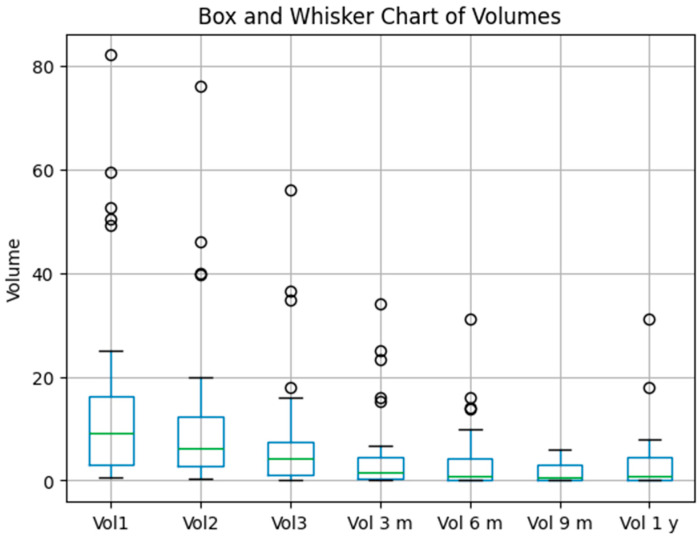
Box and whisker chart of tumor volumes at different time points.

**Figure 13 biomedicines-13-00423-f013:**
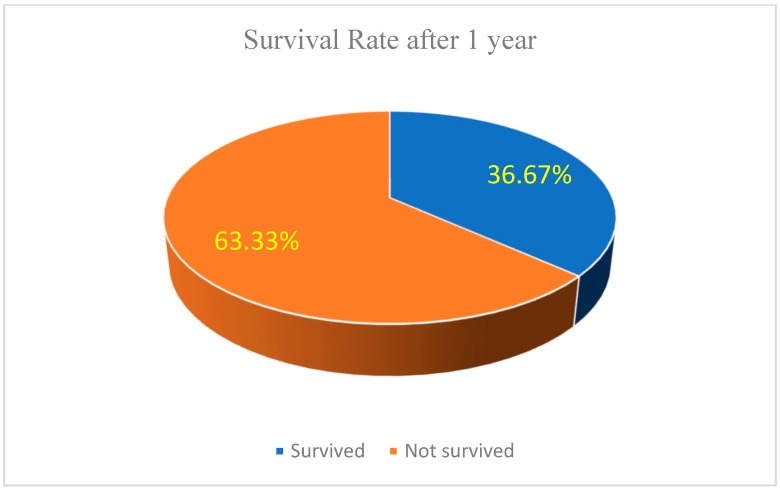
Survival rate of the patients after 1 year.

**Figure 14 biomedicines-13-00423-f014:**
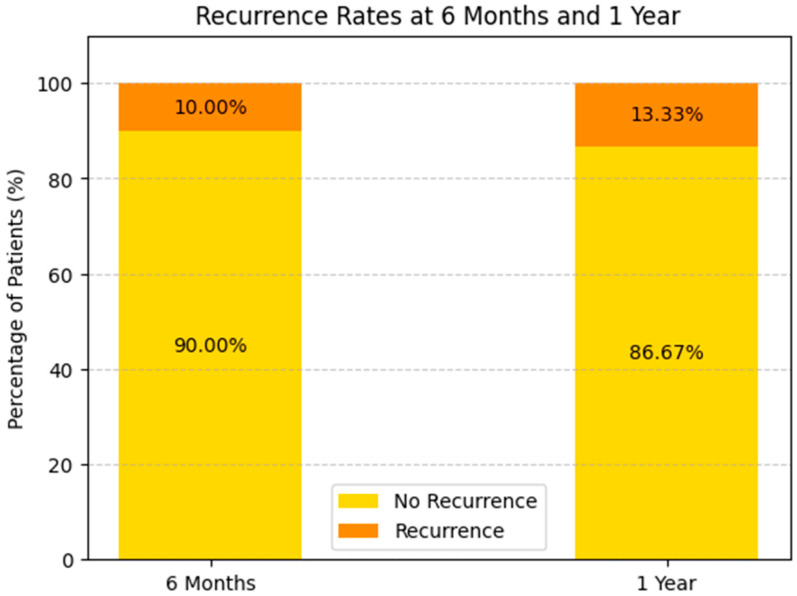
Recurrence rate at 6 months and 1 year of the study population.

**Table 1 biomedicines-13-00423-t001:** Comparative performance metrics and training times for different models.

Model	Accuracy (95% CI)	Precision (95% CI)	Recall (95% CI)	F1-Score (95% CI)	AUC (95% CI)	Total Time (s)
AlexNet	99.53 (98.90–100.00)	99.01 (97.37–100.00)	99.50 (98.42–100.00)	99.26 (98.30–100.00)	0.9998 (0.9995–1.0000)	427.04
AlexNet (TL)	100.00 (100.00–100.00)	100.00 (100.00–100.00)	100.00 (100.00–100.00)	100.00 (100.00–100.00)	1.0000 (1.0000–1.0000)	420.12
ResNet-50	100.00 (100.00–100.00)	100.00 (100.00–100.00)	100.00 (100.00–100.00)	100.00 (100.00–100.00)	1.0000 (1.0000–1.0000)	443.38
EfficientNet-B0	100.00 (100.00–100.00)	100.00 (100.00–100.00)	100.00 (100.00–100.00)	100.00 (100.00–100.00)	1.0000 (1.0000–1.0000)	439.87

**Table 2 biomedicines-13-00423-t002:** Comparison of deep learning models for brain tumor classification and prediction.

Study	Methodology	Accuracy	AUC	Key Findings	Reference
This Study	TL-enhanced AlexNet for BM classification post-GKRS	100%	1.00	Achieved perfect precision, recall, and F1-score across all classes.	--
R. Anita Jasmine & P. Arockia Jansi Rani (2020)	AlexNet + SVM for meningioma, glioma, and pituitary tumors	95%	N/A	Demonstrated strong classification performance across three tumor types.	[30]
Kapadnis and Aboli (2021)	AlexNet + CNN for tumor detection	98%	N/A	Effectively separated healthy and cancerous tissues.	[31]
Buzea et al. (2023)	AI-based analysis of metastases post-GKRS	91%	0.94	Accurately predicted early response to radiosurgery.	[32]
Azhagiri & Rajesh (2024)	Enhanced AlexNet (EAN) for brain tumors	98.6%	N/A	Outperformed standard AlexNet with enhanced performance.	[33]
Zhuodiao Kuang (2022)	Hybrid AlexNet-DCNN-ResNet for tumors	96.4%	0.98	Achieved high F1-score for distinguishing benign vs. malignant cases.	[34]
Sunita Kulkarni & Sundari (2021)	Deep CNNs with TL for MRI classification	97.2%	N/A	Demonstrated strong transfer learning-based CNN classification.	[35]
S. Nidaan Khofiya et al. (2022)	AlexNet for gliomas, meningiomas, and healthy tissues	93.8%	N/A	Achieved high classification accuracy.	[36]
B. Taşci (2021)	AlexNet-based MRI classification	92.3%	N/A	Effective classification but requires improvements.	[37]

## Data Availability

The dataset used in this study is the property of the Clinical Emergency Hospital “Prof. Dr. Nicolae Oblu” in Iasi and is hosted on Google Cloud. Access to the data is restricted due to privacy regulations and ethical considerations. Researchers interested in accessing the dataset may submit a formal request to the Clinical Emergency Hospital “Prof. Dr. Nicolae Oblu” in Iasi at Gamma Knife Department (gamma.oblu@gmail.com). Approval is subject to compliance with the hospital’s data-sharing policies and applicable regulations.

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
