# Peer review of "Streamlit Application and Deep Learning Model for Brain Metastasis Monitoring After Gamma Knife Treatment"

_biomedicines, 2025, doi:10.3390/biomedicines13020423_

Round 1
Reviewer 1 Report
Comments and Suggestions for Authors
This manuscript presents a comprehensive study that utilizes transfer learning to analyze MRI scans from 60 patients using an AlexNet-based deep learning model. The study's integration of dynamic class weighting, data augmentation, and performance evaluation metrics such as accuracy, precision, recall, F1-scores, and AUC provides some insights. Additionally, the incorporation of AI predictions into a Streamlit-based application ensures both clinical usability and accessibility. From an academic and clinical standpoint, the study is of importance and interest. However, the explanation for the choice of AlexNet as the architecture for medical image classification requires more detail. Below are my suggestions.
1. While the author briefly mentions using the AlexNet architecture, the explanation for this choice is not sufficiently clear or comprehensive. I recommend that the author consider adding a section that briefly introduces several other CNN-based models, such as VGG16, VGG19, ResNet, Inception, EfficientNet, and MobileNet, and discuss their respective strengths and weaknesses in the context of medical image classification.
Furthermore, it is essential to elaborate on why AlexNet was chosen, particularly in the context of clinical medical settings where computational resources may be limited, and rapid diagnostic results are critical. Specifically, the paper should discuss how AlexNet's computational efficiency makes it a good choice for real-time use in clinical environments.
While AlexNet has been widely applied in various domains and offers computational efficiency, other models like EfficientNet and MobileNet may offer even better prospects. EfficientNet achieves more accurate results with slightly higher computational resource consumption, making it suitable for handling larger datasets, which could be advantageous in medical image classification tasks involving more complex datasets. On the other hand, MobileNet is highly efficient with lower computational overhead, making it ideal for resource-constrained environments, such as mobile medical devices, while maintaining good classification accuracy.
In my view, the EfficientNet model shows a lot of promise for medical image analysis because of its optimized balance between accuracy and resource consumption. Additionally, from a long-term perspective, the integration of mobile detection devices in clinical settings is becoming increasingly prevalent. Thus, the use of MobileNet could provide substantial practical benefits, especially in resource-constrained environments.
I hope the author will consider focusing more on models like EfficientNet and MobileNet in future research, as these architectures offer the potential for further advancements in medical image classification while better meeting the practical limitations of clinical environments.
2. In addition to CNN-based models, Transformer-based architectures (e.g., Vision Transformer (ViT), Data-Efficient Image Transformer (DeiT), and Pooling-based Vision Transformer (PiT)) have gained significant attention in the field of medical image classification. It is recommended that the author briefly introduce these Transformer-based models in the introduction section, along with their respective strengths and weaknesses in the context of medical image classification. This would provide a more comprehensive overview of current trends in deep learning for medical imaging.
3. Section 3.4, "Comparative Analysis of GKRS Outcomes and Predictive Modeling," is rich in content and spans 9 pages (pages 18-26), accounting for one-third of the total research content. However, this section is barely mentioned in the abstract and lacks the necessary transitions in the main text, which makes it feel somewhat abrupt.
To improve the flow, I recommend the following:
- (1) The author should add 1-2 sentences in the abstract to briefly highlight the importance and key findings of Section 3.4.
- (2) The author should include 1-2 paragraphs at the end of Section 3.3 or the beginning of Section 3.4 to provide a smooth transition between the sections, making the overall structure of the paper more coherent.
4. The format of Table 2, "Comparison of Deep Learning Models for Brain Tumor Classification and Prediction," could be improved for better readability and visual appeal. It is suggested that the author revise the table to make it more concise and better organized. A cleaner and more aesthetically pleasing format would improve the clarity and presentation of the information.
Author Response
Dear Reviewer,
Thank you for your valuable comments and suggestions. We have carefully addressed each of your concerns in the attached Word file.
We appreciate your time and insights, which have significantly improved our manuscript.
Sincerely,
Prof. Dr. Calin Gh Buzea

Reviewer 2 Report
Comments and Suggestions for Authors
Overstatement of Performance: The claim of "flawless classification accuracy (100%)" (Lines 27–28) needs to be re-evaluated. It is highly unlikely that a deep learning model achieves perfect accuracy. Consider discussing potential biases, sample size limitations, and overfitting concerns.
Statistical Reporting Issues: The model’s reported performance metrics (Lines 28–30) lack critical details, such as standard deviations or confidence intervals. Suggest adding statistical significance testing or external validation results.
Clinical Utility Needs More Explanation: While the Streamlit application is introduced (Line 18), there is little information on how this tool enhances decision-making. Consider adding a brief statement on user feedback or pilot clinical testing.
Grad-CAM Visualization Mention (Lines 40–41): This is a strong claim about interpretability. Consider adding a supporting sentence on validation with expert radiologists.
Lack of Citations for AI in Medical Imaging (Lines 50–53): The manuscript states that AI has revolutionized medical imaging but does not provide key references to support this claim. Add citations from recent systematic reviews on AI in radiology.
Subjective Language (Lines 72–73): The phrase "growing demand for advanced tools" is vague. Consider quantifying the extent of clinical demand with references.
Underexplored AI Deployment Challenges (Lines 74–76): The discussion on AI deployment barriers is too brief. Consider expanding on factors such as dataset heterogeneity, explainability, and regulatory compliance.
Small Sample Size (Lines 150–154): The dataset includes only 60 patients. This is a significant limitation, as deep learning models require large datasets for generalization. A discussion on data augmentation or external dataset validation is necessary.
Inclusion Criteria Ambiguities (Lines 155–159): It is unclear whether patients with recurrent metastases or previous treatments were included. Explicitly mention whether prior chemotherapy or immunotherapy was a factor.
MRI Preprocessing Needs More Detail (Lines 214–236): The manuscript states that "images were anonymized, preprocessed, and reviewed for quality" (Line 233). What preprocessing steps were taken? Were intensity normalization or motion correction applied?
Transfer Learning Justification (Lines 317–342): AlexNet was chosen due to computational efficiency, but it is an older architecture. Why were more modern networks (e.g., Vision Transformers) not explored?
Model Overfitting Concerns (Lines 509–517): The transfer learning model achieved 100% accuracy, which is highly suspicious. Was k-fold cross-validation performed? If not, it should be included to verify generalizability.
Lack of External Validation (Lines 518–523): The study only tests performance on internal data. This raises concerns about real-world applicability. Was external data from another institution tested?
Unclear Tumor Volume Analysis Method (Lines 600–618): The manuscript claims to have identified tumor progression predictors but does not specify the methodology. Was a survival model, such as Cox regression, used?
Survival and Recurrence Analysis (Lines 652–705): The survival rate comparison lacks Kaplan-Meier survival curves. Consider adding a statistical comparison (log-rank test) between patient groups.
Comparison with Prior Studies Needs Improvement (Lines 859–862): The manuscript compares results with past studies but does not discuss why differences in accuracy occur. Did the higher accuracy result from better data preprocessing or a different dataset?
Explainability Aspects Require More Detail (Lines 858–861): The use of Grad-CAM is mentioned, but how well did it align with radiologist interpretations? A figure comparing AI heatmaps with expert annotations would strengthen this claim.
Future Work Section Needs More Depth (Lines 118–138): The manuscript suggests expanding datasets and incorporating multimodal imaging. A discussion on integrating clinical metadata (e.g., genetic markers) would be valuable.
Figure 5 (Training Loss and Accuracy Curves): The curves suggest perfect convergence, which is unrealistic for deep learning models. Was early stopping used?
Figure 10 (Confusion Matrix): It shows no misclassifications, which raises concerns about data leakage. Was the model tested on completely unseen data?
Table 1 (Comparison of Models): It compares AlexNet, ResNet-50, and EfficientNet-B0, but all have 100% accuracy. This suggests either dataset leakage or an overly simple classification problem.
Author Response
Dear Reviewer,
Thank you for your valuable comments and suggestions. We have carefully addressed each of your concerns in the attached Word file.
We appreciate your time and insights, which have significantly improved our manuscript.
Sincerely,
Prof. Dr. Calin Buzea

Round 2
Reviewer 2 Report
Comments and Suggestions for Authors
I thank the authors for addressing the concerns raised in the report. The current form of this manuscript may be considered for the publication.